# Richer Bayesian Last Layers with Subsampled NTK Features

**Sergio Calvo-Ordoñez** [* 1 2 3]   **Jonathan Plenk** [* 1 2]   **Richard Bergna** [4]   **Álvaro Cartea** [1 2]
**Yarin Gal** [3]   **Jose Miguel Hernández-Lobato** [4]   **Kamil Ciosek** [5]

## Abstract

Bayesian Last Layers (BLLs) provide a convenient and computationally efficient way to estimate uncertainty in neural networks. However, they underestimate epistemic uncertainty because they apply a Bayesian treatment only to the final layer, ignoring uncertainty induced by earlier layers. We propose a method that improves BLLs by leveraging a projection of Neural Tangent Kernel (NTK) features onto the space spanned by the last-layer features. This enables posterior inference that accounts for variability of the full network while retaining the low computational cost of inference of a standard BLL. We show that our method yields posterior variances that are provably greater or equal to those of a standard BLL, correcting its tendency to underestimate epistemic uncertainty. To further reduce computational cost, we introduce a uniform subsampling scheme for estimating the projection matrix and for posterior inference. We derive approximation bounds for both types of subsampling. Empirical evaluations on UCI regression, contextual bandits, image classification, and out-of-distribution detection tasks in image and tabular datasets, demonstrate improved calibration and uncertainty estimates compared to standard BLLs and competitive baselines, while reducing computational cost.

## 1. Introduction

Uncertainty estimates play an important role in the deployment of neural networks when decisions depend on model confidence or when there are data distribution shifts (Kendall & Gal, 2017). Several Bayesian and ensemble methods have been proposed to improve uncertainty quantification in deep learning (Huang et al., 2017; Lakshminarayanan et al., 2017; Bergna et al., 2025; He et al., 2020), but these approaches often incur substantial computational overhead relative to standard training and inference. For instance, variational methods (Blundell et al., 2015), Markov Chain Monte Carlo (Papamarkou et al., 2022; Izmailov et al., 2021), and sampling-based approximations (Maddox et al., 2019; Chen et al., 2014) introduce additional optimization or sampling loops, while techniques such as Bayesian dropout (Gal & Ghahramani, 2016) require repeated forward passes. Even single-pass methods demand architectural changes or specialized loss functions, making them difficult to apply in large models (Van Amersfoort et al., 2020; Liu et al., 2020).

Epistemic uncertainty arises from limited data, and in neural networks is commonly modeled through uncertainty over model parameters (Gal et al., 2016). The Neural Tangent Kernel (NTK) (Jacot et al., 2018) captures this effect for wide neural networks and provides a theoretically grounded way to model epistemic variability induced by weight uncertainty (Wilson et al., 2025) through the NTK Gaussian Process (NTK-GP) framework (He et al., 2020; Ordoñez et al., 2026). However, computing the NTK-GP posterior requires solving a linear system whose dimension grows with the number of training points (or total number of parameters), making it prohibitive in large-scale settings.

A common practical alternative for uncertainty estimation in neural networks is to place a Bayesian linear model on top of the final hidden representation, known as a Bayesian last layer (BLL) (Williams & Rasmussen, 2006). BLLs are computationally attractive because posterior inference reduces to Bayesian linear regression in the last-layer feature space, avoiding kernel computations over training data. In the infinite width limit, Bayesian linear regression in these features corresponds to a Gaussian process with the Neural Network Gaussian Process (NNGP) kernel (Matthews et al., 2018). However, BLLs ignore uncertainty induced by earlier layers, and therefore underestimate epistemic uncertainty compared to the NTK-GP.

Several works have attempted to address the underestimation of epistemic uncertainty in BLLs by refining the last-

---

[*]Equal contribution  [1]Mathematical Institute, University of Oxford [2]Oxford-Man Institute, University of Oxford [3]OATML, University of Oxford [4]Department of Engineering, University of Cambridge [5]Spotify. Correspondence to: Sergio Calvo-Ordoñez <sergio.calvoordonez@maths.ox.ac.uk>, Jonathan Plenk <jonathan.plenk@maths.ox.ac.uk>.

*Proceedings of the 43rd International Conference on Machine Learning*, Seoul, South Korea. PMLR 306, 2026. Copyright 2026 by the author(s).

Full NTK-GP

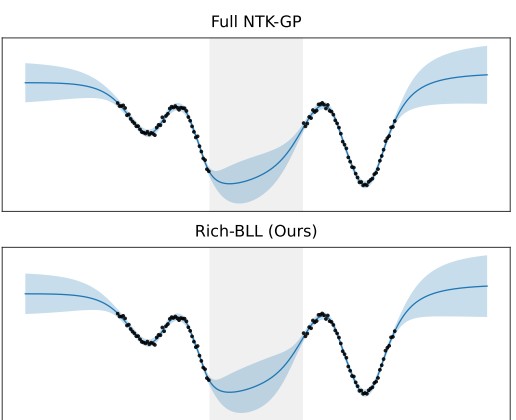

NNGP (Bayesian Last Layer)

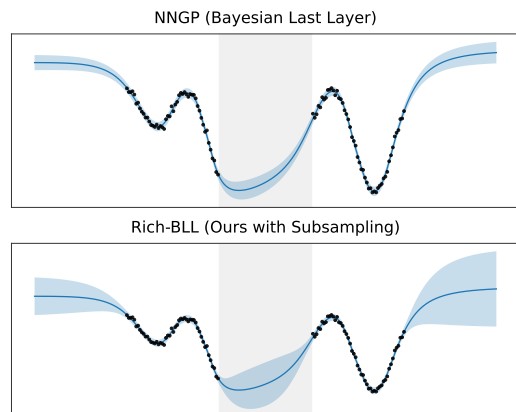

*Figure 1.* Predictive uncertainty comparison for a 1D regression problem. NNGP (BLL) underestimates epistemic uncertainty relative to the NTK-GP, while our NTK approximation (Rich-BLL) recovers richer uncertainty at BLL cost, even when using uniform subsampling.

layer posterior. Laplace-based Bayesian last-layer methods approximate the curvature of the loss around a MAP estimate to obtain a Gaussian posterior over the final layer weights, improving calibration while retaining the Bayesian treatment at the last layer (Kristiadi et al., 2020). Variational Bayesian Last Layers (VBLL) (Harrison et al., 2024) similarly restrict variational inference to the final layer, yielding a sampling-free, single-forward-pass model with quadratic complexity in the last-layer width. Other approaches extend Laplace approximations beyond the last layer using structured curvature estimates, such as Kronecker-factored approximations of the Hessian or Fisher matrix (Ritter et al., 2018), but their computational cost scales with the number of parameters. As a result, existing approaches either refine the last-layer posterior while ignoring uncertainty induced by earlier layers, or extend beyond the last layer at a computational cost that is impractical for large networks. Other approaches introduce alternative deterministic uncertainty estimators that modify the training objective or output parameterization (Mukhoti et al., 2023).

*Table 1.* Computational complexity of NTK-GP inference, NNGP, and our method with subsampling (S) and without. $N$ is the number of training points, $p$ the total number of network parameters (eNTK feature dimension), $r$ the last-layer feature dimension, and $k$ the number of subsampled training points.

| Method | Features | Time | Memory |
|---|---|---|---|
| NTK-GP/LLA | $\phi^p(x)$ | $\mathcal{O}(N^3 + N^2 p)$ | $\mathcal{O}(N^2)$ |
| NNGP/LL-LLA/BLL | $\phi^r(x)$ | $\mathcal{O}(r^3 + Nr^2)$ | $\mathcal{O}(r^2)$ |
| Rich-BLL | $L^\top \phi^r(x)$ | $\mathcal{O}(r^3 + Nr^2)$ | $\mathcal{O}(r^2)$ |
| Rich-BLL (S) | $L_S^\top \phi^r(x)$ | $\mathcal{O}(r^3 + kr^2)$ | $\mathcal{O}(r^2)$ |

In this work, we propose to approximate NTK-GP inference by modifying the kernel used in the last layer to obtain better epistemic uncertainty at the computational cost of inference in a BLL. Using NTK features from earlier layers and last-layer (NNGP) features on the training data, we estimate a small positive-definite matrix that captures the

contributions of earlier layers to the NTK, and reparameterize it via a Cholesky factor. We then obtain transformed features and perform Bayesian linear regression in this new feature space, which implicitly incorporates additional NTK structure while preserving the computational complexity of a standard BLL. To further reduce the cost, we estimate the required feature covariances using only a uniformly subsampled subset of training points and show that the resulting posterior is a sensible approximation whose error decreases with the number of subsamples. The main approximation behind our method is that the contribution of earlier-layer eNTK features can be captured by a low-dimensional correction in the last-layer feature space.

Our contributions are summarized as follows:

- We introduce a scalable approximation to NTK-GP inference that enriches Bayesian last layers by incorporating contributions from earlier layers through a low-dimensional kernel correction, providing better calibrated epistemic uncertainty at the computational cost of a standard BLL.

- We propose a uniform subsampling scheme for estimating the kernel correction and computing the posterior, enabling inference to scale with the number of subsamples rather than the full dataset size.

- We provide theoretical guarantees for our proposed method. In particular, we show that, without subsampling, the posterior variance is always more conservative than that of a standard BLL, and we derive approximation bounds for both the kernel correction and the subsampled posterior.

- We validate the proposed method in terms of calibration and uncertainty estimation on UCI regression, contextual bandits, and out-of-distribution detection tasks, while maintaining a low computational cost.

| MAP (Backbone) | NNGP (Bayesian Last Layer) | Rich-BLL (Ours) | Rich-BLL (Ours, Subsampling) |
|---|---|---|---|

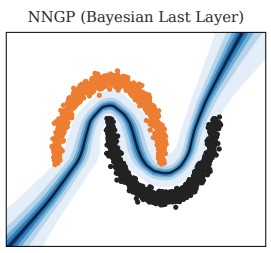 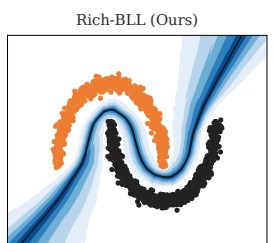 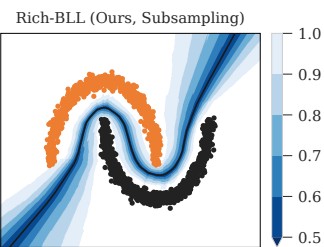

*Figure 2.* Predictive mean and uncertainty for a 1D classification toy problem. From left to right: deterministic MAP backbone, Bayesian last layer (NNGP), Rich-BLL (ours), and Rich-BLL subsampling. Shaded regions indicate predictive uncertainty.

## 2. Preliminaries

Consider a supervised regression setting in which observations are generated as $y = f_\theta(x) + \varepsilon$, with $\varepsilon \sim \mathcal{N}(0, \sigma^2 I)$ and $\sigma^2 > 0$. We study a neural network $f_\theta(x)$ with input $x \in \mathbb{R}^d$ and parameters $\theta \in \mathbb{R}^p$. Consider having trained the neural network to optimal parameters $\hat{\theta}$. A first-order Taylor expansion of the network around $\hat{\theta}$ motivates representing the effect of small parameter perturbations through the network's parameter gradients. This leads to the empirical NTK (eNTK) features, defined for a finite-width network and evaluated at $\hat{\theta}$. The eNTK features are given by the parameter-gradient

$$\phi^p(x) := \nabla_\theta f_{\hat{\theta}}(x) \in \mathbb{R}^p. \tag{1}$$

In this work, we discuss performing uncertainty quantification using a Gaussian process (GP, Williams & Rasmussen (2006)) with eNTK features. When $\hat{\theta}$ is the MAP estimate, this is equivalent to the Linearized Laplace Approximation (Daxberger et al., 2021; Immer et al., 2021). Consider training points $\mathbf{x}_1, \ldots, \mathbf{x}_N \in \mathbb{R}^d$. Denote the feature matrix by $\Phi_\mathbf{x}^p \in \mathbb{R}^{N \times p}$, and for any $\mathbf{x}_1', \ldots \mathbf{x}_{N'}' \in \mathbb{R}^d$, define the eNTK kernel matrix

$$k_{\mathbf{x},\mathbf{x}'}^p := \Phi_\mathbf{x}^p \Phi_{\mathbf{x}'}^{p\top} \in \mathbb{R}^{N \times N'}. \tag{2}$$

The posterior-predictive covariance is

$$S_{\mathbf{x}',\mathbf{x}'}^{\mathrm{ntk}} = k_{\mathbf{x}',\mathbf{x}'}^p - k_{\mathbf{x}',\mathbf{x}}^p \left( k_{\mathbf{x},\mathbf{x}}^p + \sigma^2 I_N \right)^{-1} k_{\mathbf{x},\mathbf{x}'}^p. \tag{3}$$

For a single point $x \in \mathbb{R}^d$, this gives the predictive distribution

$$p(y|x, D) = \mathcal{N}(y; f_{\hat{\theta}}(x), S^{\mathrm{ntk}}(x, x) + \sigma^2). \tag{4}$$

Similarly, the Bayesian Last Layer (BLL) kernel matrix $k_{\mathbf{x},\mathbf{x}'}^r := \Phi_\mathbf{x}^r \Phi_{\mathbf{x}'}^{r\top}$ gives the predictive uncertainty $S_{\mathbf{x}',\mathbf{x}'}^{\mathrm{bll}}$, where $r$ is the last-layer feature dimension.

## 3. Methodology

In this section, we introduce the proposed method. The idea is based on approximating the empirical NTK feature map by a low-dimensional projection onto the span of last-layer features, allowing posterior inference to be carried out entirely in the last-layer feature space. We first review the empirical NTK and its decomposition into last-layer (NNGP) and non-last-layer components. We then introduce a data-driven transformation that approximates the full NTK features using last-layer features only, and show how this leads to an efficient closed-form posterior covariance via a modified Bayesian last layer. A detailed motivation and error analysis for the low-rank projection used below is given in Appendix B. We prove that the resulting predictive uncertainty is always more conservative than that of a standard BLL. Finally, we introduce a uniform subsampling scheme that further reduces computational cost and provide theoretical guarantees on the resulting approximation error. See Table 1 for a summary of the complexities and features notation.

### 3.1. Empirical Neural Tangent Kernel Features

We begin by reviewing the empirical Neural Tangent Kernel (eNTK) representation induced by a trained neural network and its relationship to BLLs. This representation makes explicit which parameters contribute to predictive uncertainty and will allow us to identify a tractable approximation. We take the last layer to be linear, with weights and biases $\theta^{l+1} \in \mathbb{R}^r$. Denote the remaining parameters by $\theta^{\leq l} \in \mathbb{R}^m$, such that $\theta = \mathrm{vec}(\theta^{\leq l}, \theta^{l+1}) \in \mathbb{R}^{m+r} = \mathbb{R}^p$. Define the NNGP-features

$$\phi^r(x) := \nabla_{\theta^{l+1}} f_{\hat{\theta}}(x) = \mathrm{vec}(a_{\hat{\theta}}^l(x), 1) \in \mathbb{R}^r. \tag{5}$$

These are equal to the post-activation $a_{\hat{\theta}}^l(x)$ of the last hidden layer, augmented with a bias term, and form the representation used in Bayesian last layer (BLL) methods (Snoek et al., 2015). We have

$$\phi^p(x) = \mathrm{vec}\left(\phi^m(x), \phi^r(x)\right) \in \mathbb{R}^{m+r}, \tag{6}$$

where $\phi^m(x) \in \mathbb{R}^m$ is the gradient with respect to the remaining parameters. The full eNTK feature map therefore decomposes into a contribution from the last layer, $\phi^r(x)$, and a contribution from all earlier layers $\phi^m(x)$. Our method is built on approximating the contribution of $\phi^m(x)$

to the NTK using a linear transformation of the last-layer features $\phi^r(x)$.

Direct NTK-GP inference requires operating in the full parameter-gradient feature space of dimension $p = m + r$, which is computationally infeasible in modern networks. However, prior work suggests that the eigenvalues of the NTK can decay rapidly under certain conditions, leading to an effective low-dimensional structure in practice. For example, Belfer et al. (2024) analyze the spectrum of the NTK for deep residual networks and show that, under a spherical data distribution, the NTK eigenvalues decay polynomially with the index of the corresponding eigenfunctions, implying that many directions contribute weakly to the kernel. Power-series characterizations of the NTK also show that activation function properties influence the decay of kernel coefficients (Murray et al., 2022), and empirical observations find that NTK matrices often exhibit a few large eigenvalues followed by smaller ones (Bowman, 2023). Motivated by this, we seek a linear approximation of the non-last-layer features in terms of the last-layer features. Appendix B makes this motivation precise: spectral decay supports the presence of a dominant low-dimensional eNTK subspace, while the quasi-low-rank residual measures the extent to which this subspace is captured by the last-layer feature span.

## 3.2. Approximating the eNTK features for the posterior covariance

Assume there is a matrix $A \in \mathbb{R}^{m \times r}$, such that

$$\phi^m(x) \approx A\phi^r(x). \tag{7}$$

We empirically estimate $A$ by minimizing

$$\min_A \sum_{i=1}^{N} \|\phi^m(\mathbf{x}_i) - A\phi^r(\mathbf{x}_i)\|_2^2. \tag{8}$$

This corresponds to projecting the non-last-layer NTK features onto the span of the last-layer features using training data. For $N \geq r$, $\Phi_{\mathbf{x}}^{r\top}\Phi_{\mathbf{x}}^r \in \mathbb{R}^{r \times r}$ has full rank, so

$$A = \Phi_{\mathbf{x}}^{m\top}\Phi_{\mathbf{x}}^r(\Phi_{\mathbf{x}}^{r\top}\Phi_{\mathbf{x}}^r)^{-1} \in \mathbb{R}^{m \times r}. \tag{9}$$

Now, by defining

$$B := \begin{pmatrix} A \\ I_r \end{pmatrix} \in \mathbb{R}^{(m+r) \times r}, \tag{10}$$

we can express the full NTK feature map using only $r$-dimensional features. In particular, the approximation

$$\phi^p(x) \approx B\phi^r(x) =: \phi^B(x) \in \mathbb{R}^{m+r}, \tag{11}$$

i.e., $\Phi_{\mathbf{x}}^p \approx \Phi_{\mathbf{x}}^r B^\top$, implies that the NTK-GP inference can be carried out using a modified Bayesian linear model in the last-layer feature space. Define the approximate kernel

$k_{\mathbf{x},\mathbf{x}'}^B := \Phi_{\mathbf{x}}^r B^\top B \Phi_{\mathbf{x}'}^{r\top} \in \mathbb{R}^{N \times N'}$. Then the approximate posterior predictive covariance is

$$S_{\mathbf{x}',\mathbf{x}'}^B := k_{\mathbf{x}',\mathbf{x}'}^B - k_{\mathbf{x}',\mathbf{x}}^B \left(k_{\mathbf{x},\mathbf{x}}^B + \sigma^2 I_N\right)^{-1} k_{\mathbf{x},\mathbf{x}'}^B. \tag{12}$$

This involves inverting an $N \times N$ matrix. After applying Woodbury's Lemma, we would still require inverting an $p \times p$ matrix. However, using the special structure of the feature map, we can reduce this to inverting an $r \times r$ matrix:

**Theorem 3.1.** *Let $B \in \mathbb{R}^{(m+r) \times r}$ have full column rank. Define $\phi^B(x) := B\phi^r(x)$. Then*

$$S_{\mathbf{x}',\mathbf{x}'}^B = \Phi_{\mathbf{x}'}^r \left(\frac{1}{\sigma^2}\Phi_{\mathbf{x}}^{r\top}\Phi_{\mathbf{x}}^r + (B^\top B)^{-1}\right)^{-1} \Phi_{\mathbf{x}'}^{r\top}. \tag{13}$$

The proof is in Appendix A.1.

Moreover, consider the Cholesky decomposition $B^\top B = LL^\top$ with lower triangular $L \in \mathbb{R}^{r \times r}$. Defining the lower-dimensional features

$$\phi^L(x) := L^\top \phi^r(x) \in \mathbb{R}^r \tag{14}$$

further allows us to simplify the predictive covariance:

**Theorem 3.2.** *Using the features $\phi^L(x) \in \mathbb{R}^r$ is equivalent to using $\phi^B(x) \in \mathbb{R}^{m+r}$, and the predictive covariance can be written as*

$$S_{\mathbf{x}',\mathbf{x}'}^B = \Phi_{\mathbf{x}'}^L \left(\frac{1}{\sigma^2}\Phi_{\mathbf{x}}^{L\top}\Phi_{\mathbf{x}}^L + I_r\right)^{-1} \Phi_{\mathbf{x}'}^{L\top} \tag{15}$$

$$= \Phi_{\mathbf{x}'}^r L \left(\frac{1}{\sigma^2}L^\top \Phi_{\mathbf{x}}^{r\top}\Phi_{\mathbf{x}}^r L + I_r\right)^{-1} L^\top \Phi_{\mathbf{x}'}^{r\top}. \tag{16}$$

The proof is in Appendix A.1.

A key property of the proposed approximation is that it never reduces predictive uncertainty relative to a standard Bayesian last layer. As the amount of data grows, both methods concentrate and predictive uncertainty vanishes, while in finite-data or extrapolation regimes the additional NTK components induce higher uncertainty where the model is less constrained by observations. We formalize this property in the following theorem:

**Theorem 3.3.** *Using the approximation of NTK features via $x \mapsto \phi^B(x) = B\phi^r(x)$ (or, equivalently, $\phi^L(x)$) always gives higher predictive uncertainty than using the BLL features $x \mapsto \phi^r(x)$. In other words,*

$$S_{\mathbf{x}',\mathbf{x}'}^B \succeq S_{\mathbf{x}',\mathbf{x}'}^{\mathrm{bll}}. \tag{17}$$

The proof is in Appendix A.1.

For our method we only need the lower-triangular $L \in \mathbb{R}^{r \times r}$, and in particular there is no need to store $A \in \mathbb{R}^{m \times r}$.

In Appendix C we show how to compute $L$ efficiently for a large number of hidden parameters $m$.

As we have presented above, the transformation matrix $A$ is defined via a least-squares objective in expectation over the data distribution. In practice, $A$ must be estimated from finitely many training points, and the accuracy of the resulting NTK approximation depends on the quality of this estimate. The following result quantifies how well the empirical estimator $\hat{A}$ concentrates around the solution of the expected least-squares problem as the number of data points increases.

**Theorem 3.4.** *Define the true map $A \in \mathbb{R}^{m \times r}$ by*

$$\min_A \mathbb{E}_{x \sim P_X} \left[ \|\phi^m(x) - A\phi^r(x)\|_2^2 \right]. \quad (18)$$

*Consider its approximation $\hat{A}$ from $N$ data points. Assume that the train points are in a compact set and thus there is $K$ such that for all $x$, $\|\phi^m(x)\|_2 \leq K$ and $\|\phi^r(x)\|_2 \leq K$. Then, there is a constant $K'$ depending on $K$ and $\lambda_{\min}(\mathbb{E}_{x \sim P_X}[\phi^r(x)\phi^r(x)^\top]) > 0$, such that: For any $\delta > 0$, there is $N$ large enough such that with probability $1 - \delta$ over iid samples $\mathbf{x}_1, \ldots, \mathbf{x}_N \sim P_X$,*

$$\|\hat{A} - A\|_2 \leq K' \sqrt{\frac{\log\left(2(m+r)/\delta\right)}{N}}. \quad (19)$$

The proof is in Appendix A.3.

### 3.3. Subsampling Data Points

As shown in Theorem 3.4, the estimator $\hat{A}$ concentrates around the solution of the least-squares problem at rate $O(1/\sqrt{N})$. The same concentration bound applies when $\hat{A}$ is estimated from a uniformly subsampled set of size $k$, with the rate becoming dependent on the size of the subsampled set, i.e., $O(1/\sqrt{k})$. This implies that accurate estimation of the feature transformation only requires a number of samples proportional to the feature dimension $r$, rather than the full training set, and can therefore be performed efficiently using a subsample of size $k$.

Beyond estimating the feature map, our proposed method scales with $O(Nr^2 + r^3)$ instead of $O(N^3 + N^2r)$ during posterior inference, as it only requires inverting an $r \times r$ matrix involving the population matrix

$$\Phi_{\mathbf{x}}^{L\top}\Phi_{\mathbf{x}}^L = \sum_{i=1}^N \phi^L(\mathbf{x}_i)\phi^L(\mathbf{x}_i)^\top =: N\hat{\Sigma}_N \in \mathbb{R}^{r \times r}. \quad (20)$$

For large $N$ this may still be prohibitively expensive. In this section we present a subsampling approach which approximates the population matrix with

$$\Phi_{\mathbf{x}}^{L\top}\Phi_{\mathbf{x}}^L \approx \frac{N}{k} \sum_{i=1}^k \phi^L(\mathbf{x}_{s_i})\phi^L(\mathbf{x}_{s_i})^\top =: N\hat{\Sigma}_k \in \mathbb{R}^{r \times r}. \quad (21)$$

The indices $s_1, \ldots, s_k \in \{1, \ldots, N\}$ are drawn uniformly without replacement. This gives the subsampled estimate of predictive uncertainty

$$S_{\mathbf{x}',\mathbf{x}'}^{B,k} := \Phi_{\mathbf{x}'}^L \left( \frac{1}{\sigma^2}N\hat{\Sigma}_k + I_r \right)^{-1} \Phi_{\mathbf{x}'}^{L\top}. \quad (22)$$

Since the posterior depends on an $r \times r$ covariance matrix, it is natural to expect that only $O(r)$ observations are required for accurate estimation. We formally prove this intuition using a matrix concentration bound:

**Theorem 3.5.** *Assume that the training and test points are in a compact set, and the feature vector $\phi^L(x) = L^\top \phi^r(x) \in \mathbb{R}^r$ is bounded: For all $x$, $\|\phi^L(x)\|_2 \leq K$. Assume that the smallest eigenvalue of $\Sigma := \mathbb{E}_{x \sim P_X}\left[\phi^L(x)\phi^L(x)^\top\right] \in \mathbb{R}^{r \times r}$ is positive. Let $N \geq k \geq \frac{8}{3}\log\left(4r/\delta\right)\left(\frac{8K^2}{\lambda_{\min}(\Sigma)}\right)^2$. Then with probability of at least $1 - \delta$ over iid samples $\mathbf{x}_1, \ldots, \mathbf{x}_N \sim P_X$: For any $N'$ test points $\mathbf{x}'_1, \ldots, \mathbf{x}'_{N'}$,*

$$\|S_{\mathbf{x}',\mathbf{x}'}^{B,k} - S_{\mathbf{x}',\mathbf{x}'}^B\|_2 \leq N' \frac{2K^4}{\lambda_{\min}(\Sigma)}\sqrt{\frac{8\log(4r/\delta)}{3k}}. \quad (23)$$

In particular for a single test point, i.e., $N' = 1$, we get a bound on the approximation of its predictive variance. We note that this bound is remarkable, as it does not grow with the number of training points $N$ used for the matrix $S_{\mathbf{x}',\mathbf{x}'}^B$ that we are trying to estimate. The proof is in Appendix A.4, and we prove an alternative bound in Appendix A.5. Note that the smallest eigenvalue of the population matrix $\Sigma$ can be written as

$$\lambda_{\min}(\Sigma) = \min_{\|v\|_2 = 1} \mathbb{E}_{x \sim P_X}[(v^\top \phi^L(x))^2]. \quad (24)$$

Thus it measures how close the features are to being linearly dependent on the data distribution, and it is positive if and only if there is no $v \neq 0$ such that

$$\mathbb{P}_{x \sim P_X}(v^\top \phi^L(x) = 0) = 1. \quad (25)$$

## 4. Experiments

We evaluate our proposed method across a range of settings designed to assess uncertainty quality. Experiments cover illustrative toy examples (Figures 1 and 2), supervised regression, contextual bandits, and out-of-distribution detection, allowing us to evaluate predictive calibration, decision-making performance under uncertainty, and robustness to distribution shift. Across all settings, we compare against standard Bayesian last layer and common uncertainty estimation baselines under standard evaluations. We refer to the method introduced in Section 3.2 as Rich-BLL, and to the subsampling variant from Section 3.3 as Rich-BLL (S).

## 4.1. Experimental Setup

Across all experiments, we use fixed data splits and evaluate multiple uncertainty estimation methods under a common training and evaluation protocol. For each dataset and random seed, we train a neural network backbone using the $\mu$-parameterization ($\mu$P) (Yang & Hu, 2020), which ensures width-stable training and good kernel behavior, and is used in large-scale neural network training (Hu et al., 2022). When applicable, uncertainty baselines that rely on the same backbone (e.g., BLLs and NTK/Laplace-based methods) use the identical $\mu$P-trained network. Methods that admit a post-hoc formulation compute the predictive uncertainty on top of the frozen backbone via GP inference in feature space, while other baselines (e.g., dropout and sampling-based methods) rely on different training procedures to incorporate uncertainty.

For supervised regression experiments (Section 4.2), following Harrison et al. (2024), we use fixed train/validation/test splits generated with a fixed random seed and shared across all methods. Inputs are standardized using the training set mean and standard deviation, and targets are centered by subtracting the mean of the training set. Results are evaluated using test Gaussian negative log-likelihood (NLL). In addition, for the *Wine Quality* dataset we consider an out-of-distribution (OOD) setting, treating red wine as in-distribution and white wine as OOD, and report AUROC to assess separability.

For the contextual bandit experiments (Section 4.3), we consider the *Wheel* Bandit environment as presented in Riquelme et al. (2018). Rewards are corrupted with Gaussian noise, and task difficulty is controlled by the parameter $\delta$. Policies are parameterized by an MLP backbone trained online on action-conditioned inputs formed by concatenating the context with a one-hot encoding of the action. Exploration is performed using Thompson sampling (Thompson, 1933), where rewards are sampled from the posterior over function values induced by each uncertainty estimation method. We report cumulative regret normalized by the cumulative regret of a uniform random policy.

For the image classification experiments (Section 4.4), we evaluate both predictive performance and uncertainty quality under distribution shift. Models are trained on in-distribution (ID) data from *CIFAR-10* and evaluated on ID test data as well as out-of-distribution (OOD) datasets. We consider *SVHN* as a soft OOD dataset and *CIFAR-100* as a hard OOD dataset. Predictive performance on ID data is measured using test accuracy, test negative log-likelihood (NLL), and expected calibration error (ECE), while OOD detection performance is evaluated using the area under the receiver operating characteristic curve (AUROC).

For all of the experiments, results are reported by averaging across multiple seeds and including the standard error. Further details on model architectures and training and evaluation hyperparameters are provided in Appendix D.

## 4.2. Regression

We assess predictive uncertainty for tabular regression on five UCI datasets: *Boston Housing, Concrete, Energy, Power*, and *Wine Quality*. We benchmark our method with subsampling (40% of the original datasets) and without against the MAP (the deterministic network), BLL/NNGP, Variational Bayesian Last Layers (VBLL), alongside standard uncertainty quantification baselines including MC dropout (Gal & Ghahramani, 2016), SWAG (Maddox et al., 2019), Bayes-by-Backprop (Blundell et al., 2015), ensembles, and a GP with the RBF kernel. We closely follow the training procedure described in Harrison et al. (2024), and report test Gaussian negative log-likelihood (NLL) as the primary metric and focus on (i) the gap in predictive uncertainty between BLL and our Rich-BLL, and (ii) the extent to which Rich-BLL matches the performance of commonly used but computationally expensive baselines.

Table 2 shows how across all datasets, Rich-BLL consistently improves over the Bayesian last layer, yielding lower test NLL while keeping the same computational cost (when using the full dataset) or even lower (when subsampling). This indicates that incorporating contributions from earlier layers leads to generally better calibrated predictive uncertainty than last layer only models in these regression tasks. Furthermore, Rich-BLL, even after applying the subsampling, performs comparably to VBLL and other single-model uncertainty baselines, and in several cases approaches the performance of more expensive methods such as ensembles.

Table 4 shows that Rich-BLL improves OOD detection performance over BLLs on the *Wine Quality* dataset, achieving higher AUROC. The subsampled variant performs comparably to the full method, indicating that the separability is robust to approximation. It is remarkable that our subsampling scheme has minimal effect on performance in these benchmarks. Rich-BLL (S) closely matches the full Rich-BLL across all datasets, with differences within standard error. This is consistent with the approximation guarantees and backs up that posterior inference can be scaled without degrading uncertainty estimates.

## 4.3. Contextual Bandit

We evaluate contextual bandits to assess actionable uncertainty, where posterior uncertainty directly affects exploration decisions. We use the *Wheel* Bandit benchmark (Riquelme et al., 2018), which is designed to stress exploration–exploitation trade-offs: the optimal action yields high reward only in a small region of the context space,

*Table 2.* Test Gaussian negative log-likelihood (NLL; lower is better) on UCI regression benchmarks. Results are averaged over 20 random seeds and reported as mean ± standard error. Rich-BLL consistently improves over Bayesian last layers (NNGP/BLL) and matches or outperforms other uncertainty baselines, with minimal degradation under subsampling (Rich-BLL (S)).

|  | *Boston* | *Concrete* | *Energy* | *Power* | *Wine* |
|---|---|---|---|---|---|
|  | NLL ($\downarrow$) | NLL ($\downarrow$) | NLL ($\downarrow$) | NLL ($\downarrow$) | NLL ($\downarrow$) |
| Rich-BLL | $\mathbf{2.61 \pm 0.04}$ | $\mathbf{3.10 \pm 0.03}$ | $0.75 \pm 0.03$ | $\mathbf{2.78 \pm 0.01}$ | $\mathbf{1.00 \pm 0.01}$ |
| Rich-BLL (S) | $2.62 \pm 0.04$ | $\mathbf{3.10 \pm 0.04}$ | $0.78 \pm 0.03$ | $\mathbf{2.78 \pm 0.01}$ | $1.01 \pm 0.01$ |
| NNGP (BLL) | $2.78 \pm 0.06$ | $3.39 \pm 0.06$ | $0.92 \pm 0.01$ | $2.82 \pm 0.01$ | $1.03 \pm 0.01$ |
| VBLL | $\mathbf{2.75 \pm 0.06}$ | $\mathbf{3.19 \pm 0.05}$ | $0.74 \pm 0.04$ | $\mathbf{2.78 \pm 0.01}$ | $1.05 \pm 0.01$ |
| MAP | $2.79 \pm 0.09$ | $4.77 \pm 0.15$ | $0.93 \pm 0.03$ | $3.28 \pm 0.02$ | $1.02 \pm 0.00$ |
| RBF GP | $2.84 \pm 0.09$ | $3.24 \pm 0.13$ | $\mathbf{0.66 \pm 0.04}$ | $2.89 \pm 0.01$ | $\mathbf{0.97 \pm 0.01}$ |
| Dropout | $2.62 \pm 0.05$ | $3.19 \pm 0.02$ | $1.80 \pm 0.03$ | $2.81 \pm 0.01$ | $1.01 \pm 0.01$ |
| Ensemble | $\mathbf{2.48 \pm 0.09}$ | $\mathbf{3.15 \pm 0.03}$ | $\mathbf{0.94 \pm 0.02}$ | $\mathbf{2.75 \pm 0.02}$ | $\mathbf{0.98 \pm 0.01}$ |
| SWAG | $2.65 \pm 0.01$ | $3.19 \pm 0.03$ | $1.20 \pm 0.06$ | $2.79 \pm 0.01$ | $1.04 \pm 0.01$ |
| BBB | $2.54 \pm 0.04$ | $2.99 \pm 0.03$ | $1.22 \pm 0.01$ | $2.77 \pm 0.01$ | $1.05 \pm 0.01$ |

*Table 3.* Cumulative regret on the *Wheel* Bandit benchmark, averaged over 10 random seeds and reported as mean ± standard error. Results are shown for increasing difficulty levels $\delta$. Lower values indicate better exploration performance. [†] indicates that the results were borrowed from (Harrison et al., 2024).

|  | $\delta = 0.5$ | $\delta = 0.7$ | $\delta = 0.9$ | $\delta = 0.95$ | $\delta = 0.99$ |
|---|---|---|---|---|---|
| Rich-BLL | $0.48 \pm 0.01$ | $0.92 \pm 0.01$ | $2.60 \pm 0.09$ | $\mathbf{4.70 \pm 0.09}$ | $\mathbf{21.80 \pm 1.60}$ |
| Rich-BLL (S) | $0.50 \pm 0.01$ | $\mathbf{0.88 \pm 0.01}$ | $2.63 \pm 0.12$ | $4.76 \pm 0.09$ | $22.30 \pm 1.63$ |
| NNGP (BLL) | $1.20 \pm 0.03$ | $1.95 \pm 0.04$ | $5.30 \pm 0.16$ | $12.10 \pm 0.85$ | $55.80 \pm 2.30$ |
| VBLL[†] | $\mathbf{0.46 \pm 0.01}$ | $0.89 \pm 0.01$ | $\mathbf{2.54 \pm 0.02}$ | $4.82 \pm 0.03$ | $24.44 \pm 0.71$ |
| NeuralLinear[†] | $1.10 \pm 0.02$ | $1.77 \pm 0.03$ | $4.32 \pm 0.11$ | $11.42 \pm 0.97$ | $52.64 \pm 2.04$ |
| NeuralLinear-MR[†] | $0.95 \pm 0.02$ | $1.60 \pm 0.03$ | $4.65 \pm 0.18$ | $9.56 \pm 0.36$ | $49.63 \pm 2.41$ |
| LinDiagPost[†] | $1.12 \pm 0.03$ | $1.80 \pm 0.08$ | $5.06 \pm 0.14$ | $8.99 \pm 0.33$ | $37.77 \pm 2.18$ |

*Table 4.* OOD detection performance on the *Wine Quality* dataset. AUROC ($\uparrow$) for distinguishing ID samples (red wine) from OOD samples (white wine) using predictive uncertainty. Results are reported as mean ± standard error over 10 random seeds.

|  | AUROC ($\uparrow$) |
|---|---|
| Rich-BLL | $\mathbf{0.96 \pm 0.00}$ |
| Rich-BLL (S) | $\mathbf{0.96 \pm 0.00}$ |
| NNGP (BLL) | $0.88 \pm 0.01$ |
| VBLL | $0.95 \pm 0.00$ |

while safer actions provide moderate reward elsewhere. As a result, methods that underestimate epistemic uncertainty may fail to discover the optimal arm and incur high regret.

We compare Bayesian last layers (NNGP/BLL) and our Rich-BLL (with and without subsampling) to standard neural bandit baselines implemented in our codebase, including NeuralLinear and per-arm VBLL. For Rich-BLL (S) we subsample the replay buffer when forming the posterior: at each update we draw a fixed-size subset (seed-dependent), with the subset size capped by the data available and never below the feature dimension for numerical stability. For the default Rich-BLL—or with Rich-BLL (S) when the buffer is still small—the posterior uses the full data instead. For

comparison with the baselines, we used an empirical Bayes heuristic that periodically sets the aleatoric noise variance to the mean of recent squared prediction errors (over a rolling window). Performance is measured using cumulative regret normalized relative to a uniformly random policy and averaged over multiple random seeds.

Table 3 reports cumulative regret across different difficulty levels. Rich-BLL substantially outperforms the standard NNGP baseline for all values of $\delta$, with the gap widening as the problem becomes more exploration-heavy. This confirms that incorporating uncertainty contributions beyond the last layer leads to more effective exploration, particularly in regimes where the optimal action is rare and requires sustained uncertainty-driven exploration. The subsampled variant Rich-BLL (S) has a close performance to the full method across all difficulty levels, indicating that posterior approximation via subsampling is not noticeably detrimental. In contrast, last-layer and linearized baselines incur significantly higher regret in the high-$\delta$ regime, consistent with underestimation of epistemic uncertainty.

*Table 5.* Results for CIFAR-10 image classification. We report test accuracy, ECE, NLL, and AUROC for OOD detection on SVHN and CIFAR-100 (mean $\pm$ standard error).

| Method | NLL ($\downarrow$) | ECE ($\downarrow$) | SVHN AUROC ($\uparrow$) | CIFAR-100 AUROC ($\uparrow$) |
|---|---|---|---|---|
| NNGP (BLL) | $0.58 \pm 0.00$ | $0.046 \pm 0.01$ | $0.88 \pm 0.01$ | $0.62 \pm 0.02$ |
| Rich-BLL (S) | $\mathbf{0.56 \pm 0.00}$ | $0.029 \pm 0.01$ | $\mathbf{0.91 \pm 0.02}$ | $\mathbf{0.65 \pm 0.01}$ |
| LL Laplace | $0.57 \pm 0.00$ | $0.044 \pm 0.00$ | $0.84 \pm 0.01$ | $0.55 \pm 0.00$ |
| SNGP | $0.89 \pm 0.00$ | $\mathbf{0.027 \pm 0.001}$ | $0.85 \pm 0.01$ | $0.52 \pm 0.00$ |
| LL Dropout | $0.56 \pm 0.00$ | $0.031 \pm 0.002$ | $0.78 \pm 0.02$ | $0.62 \pm 0.00$ |
| Dropout | $\mathbf{0.31 \pm 0.05}$ | $\mathbf{0.013 \pm 0.003}$ | $\mathbf{0.92 \pm 0.001}$ | $0.61 \pm 0.02$ |
| Ensemble | $0.47 \pm 0.00$ | $0.030 \pm 0.002$ | $0.80 \pm 0.20$ | $\mathbf{0.65 \pm 0.01}$ |
| BBB | $1.39 \pm 0.02$ | $0.220 \pm 0.007$ | $0.89 \pm 0.01$ | $0.54 \pm 0.00$ |

## 4.4. Image Classification

For image classification, we train a convolutional neural network on *CIFAR-10* and evaluate both predictive calibration on in-distribution data and robustness under distribution shift. In-distribution performance is assessed using test NLL and ECE. For OOD detection, we use *SVHN* as a soft OOD dataset and *CIFAR-100* as a hard OOD dataset, reporting the AUROC. We considered two adapted constructions of our method for classification: estimating a class-specific feature transformation matrix, and a single global transformation shared across classes. In practice, the global transformation performed better in our experiments so we use it throughout. We used a similar procedure to the feature extraction process described in Park et al. (2023) (we provide a breakdown of the algorithm in Appendix C) to extract the eNTK features needed to estimate the transformation matrix. At test time we extract penultimate features, choose the predicted class, and use Rich-BLL inference to obtain a logit-level predictive variance, which serves as the OOD score. NNGP is the same pipeline without the transformation.

Table 5 shows that Rich-BLL improves over the NNGP/BLL baseline consistently, yielding lower NLL and ECE on (test) in-distribution data while also achieving higher AUROC on both *SVHN* and *CIFAR-100*. This indicates that incorporating additional NTK structure improves both calibration and OOD separability compared to last-layer uncertainty alone. The subsampled variant attains comparable performance, suggesting that the approximation does not degrade uncertainty quality in this setting. Compared to alternative post-hoc methods such as last-layer Laplace and SNGP, Rich-BLL provides more competitive OOD detection while maintaining strong calibration and being competitive with the rest of the baselines.

## 5. Related Work

**Linearized Laplace and NTK-based uncertainty.** A common route to uncertainty estimation in neural networks is to approximate the model locally around a trained solution via a first-order expansion, leading to linearized models

whose predictive uncertainty can be expressed in closed form (MacKay, 1992; 2003). This viewpoint is closely connected to Laplace approximations in weight space (Williams & Rasmussen, 2006; Daxberger et al., 2021) and to the generalized Gauss–Newton (GGN) approximation (Roy et al., 2024), which has been interpreted as performing inference in a locally linearized model (Martens & Grosse, 2015; Immer et al., 2021). In parallel, the Neural Tangent Kernel (NTK) characterizes the training dynamics of wide networks and motivates GP-style uncertainty computation using Jacobian/NTK features (Jacot et al., 2018; Lee et al., 2019; Arora et al., 2019). Empirical NTK feature constructions and their use for uncertainty estimation have been studied in several settings, including connections to ensemble and SGD-based approximate Bayesian inference (He et al., 2020; Wilson et al., 2025). These approaches provide a principled account of parameter-induced variability under linearization, but direct implementations can be expensive due to Jacobian construction and large linear systems.

**Scalable curvature and kernel approximations.** To make Laplace and linearization-based methods practical, many works propose structured or low-rank approximations to curvature matrices, including Kronecker-factored and related factorizations of the Fisher/GGN (Martens & Grosse, 2015; Ritter et al., 2018; Kristiadi et al., 2020), as well as diagonal, block-diagonal, and low-rank variants (Daxberger et al., 2021; Deng et al., 2022). Complementary approaches improve scalability via efficient implementations and software tooling (Weber et al., 2025). From a GP point of view, scalability is also addressed through approximations to kernel computations (e.g., inducing-point methods and sparse variational GPs) that reduce dependence on the number of training points (Titsias, 2009; Hoffman et al., 2013; Wilson & Nickisch, 2015). These methods target different bottlenecks related to curvature, Jacobians, or kernel matrices, but generally trade off computational cost against approximation fidelity. Our equation (7) and its link to the low rank of the NTK has been used in parallel work on neural network optimization (Ciosek et al., 2025); however, that work does not study uncertainty estimation.

**Post-hoc function-space uncertainty with fixed predictors.** A related line of work models predictive uncertainty directly in function space on top of a trained neural network, without relying on parameter-space linearization or Jacobian-based features. Fixed-mean GP approaches treat the network as a deterministic predictor and fit a GP model for uncertainty using variational methods (Ortega et al., 2024). More recently, activation-space methods have been proposed that attach probabilistic models to intermediate representations of frozen networks. Bergna et al. (2026) propose a replacement for deterministic activations with Gaussian process approximations whose posterior mean exactly matches the original activations, and propagates uncertainty through the network via local approximations.

## 6. Conclusion

We proposed a scalable approach to improving Bayesian last layer uncertainty estimation by approximating NTK-GP inference using a low-dimensional kernel correction. The method incorporates contributions from earlier layers while retaining the computational efficiency of standard Bayesian last layers, and admits further scalability through uniform subsampling with theoretical guarantees. Empirical results on regression, contextual bandits, image classification, and out-of-distribution detection show that the proposed approach consistently improves uncertainty over Bayesian last layers and remains competitive with more expensive uncertainty estimation methods.

## Acknowledgements

SCO's research is supported by the Oxford-Man Institute through the EPSRC Centre for Doctoral Training in Mathematics of Random Systems: Analysis, Modelling and Simulation (EPSRC Grant EP/S023925/1). JP acknowledges financial support from the Oxford-Man Institute. SCO, JP, and AC acknowledge the support of the Oxford-Man Institute for providing computational resources. YG acknowledges the funding under the Horizon Europe grant 101213369 DVPS. JMHL acknowledges funding from AI Hub in Generative Models, under grant EP/Y028805/1.

## Impact Statement

This paper studies scalable methods for uncertainty estimation in neural networks. The work is methodological in nature and does not raise new ethical or societal concerns beyond those commonly associated with machine learning research.

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

# A. Proofs

## A.1. Properties of the Approximate Predictive Uncertainty

**Theorem 3.1.** *Let $B \in \mathbb{R}^{(m+r) \times r}$ have full column rank. Define $\phi^B(x) := B\phi^r(x)$. Then*

$$S^B_{\mathbf{x}',\mathbf{x}'} = \Phi^r_{\mathbf{x}'} \left( \frac{1}{\sigma^2} \Phi^{r\top}_{\mathbf{x}} \Phi^r_{\mathbf{x}} + (B^\top B)^{-1} \right)^{-1} \Phi^{r\top}_{\mathbf{x}'}. \tag{13}$$

*Proof of Theorem 3.1.* Using Woodbury's matrix inversion Lemma we write for any $\mathbf{x}'_1, \ldots, \mathbf{x}'_{N'} \in \mathbb{R}^d$,

$$S^B_{\mathbf{x}',\mathbf{x}'} = \Phi^B_{\mathbf{x}'} \Phi^{B\top}_{\mathbf{x}'} - \Phi^B_{\mathbf{x}'} \Phi^{B\top}_{\mathbf{x}} \left( \Phi^B_{\mathbf{x}} \Phi^{B\top}_{\mathbf{x}} + \sigma^2 I_N \right)^{-1} \Phi^B_{\mathbf{x}} \Phi^{B\top}_{\mathbf{x}'} \tag{26}$$

$$= \Phi^B_{\mathbf{x}'} \left( \frac{1}{\sigma^2} \Phi^{B\top}_{\mathbf{x}} \Phi^B_{\mathbf{x}} + I_p \right)^{-1} \Phi^{B\top}_{\mathbf{x}'} \tag{27}$$

$$= \Phi^r_{\mathbf{x}'} B^\top \left( \frac{1}{\sigma^2} B \Phi^{r\top}_{\mathbf{x}} \Phi^r_{\mathbf{x}} B^\top + I_p \right)^{-1} B \Phi^{r\top}_{\mathbf{x}'}. \tag{28}$$

This still involves inverting an $p \times p$ matrix. However, due to

$$\left( \frac{1}{\sigma^2} B \Phi^{r\top}_{\mathbf{x}} \Phi^r_{\mathbf{x}} B^\top + I_p \right)^{-1} B \left( \frac{1}{\sigma^2} \Phi^{r\top}_{\mathbf{x}} \Phi^r_{\mathbf{x}} B^\top B + I_r \right) = B, \tag{29}$$

we have the following push-through identity:

$$\left( \frac{1}{\sigma^2} B \Phi^{r\top}_{\mathbf{x}} \Phi^r_{\mathbf{x}} B^\top + I_p \right)^{-1} B = B \left( \frac{1}{\sigma^2} \Phi^{r\top}_{\mathbf{x}} \Phi^r_{\mathbf{x}} B^\top B + I_r \right)^{-1}. \tag{30}$$

Hence, we only need to invert a $r \times r$ matrix:

$$S^B_{\mathbf{x}',\mathbf{x}'} = \Phi^r_{\mathbf{x}'} B^\top B \left( \frac{1}{\sigma^2} \Phi^{r\top}_{\mathbf{x}} \Phi^r_{\mathbf{x}} B^\top B + I_r \right)^{-1} \Phi^{r\top}_{\mathbf{x}'} \tag{31}$$

$$= \Phi^r_{\mathbf{x}'} \left( \left( \frac{1}{\sigma^2} \Phi^{r\top}_{\mathbf{x}} \Phi^r_{\mathbf{x}} (B^\top B) + I_r \right) (B^\top B)^{-1} \right)^{-1} \Phi^{r\top}_{\mathbf{x}'} \tag{32}$$

$$= \Phi^r_{\mathbf{x}'} \left( \frac{1}{\sigma^2} \Phi^{r\top}_{\mathbf{x}} \Phi^r_{\mathbf{x}} + (B^\top B)^{-1} \right)^{-1} \Phi^{r\top}_{\mathbf{x}'}. \tag{33}$$

$\square$

**Theorem 3.2.** *Using the features $\phi^L(x) \in \mathbb{R}^r$ is equivalent to using $\phi^B(x) \in \mathbb{R}^{m+r}$, and the predictive covariance can be written as*

$$S^B_{\mathbf{x}',\mathbf{x}'} = \Phi^L_{\mathbf{x}'} \left( \frac{1}{\sigma^2} \Phi^{L\top}_{\mathbf{x}} \Phi^L_{\mathbf{x}} + I_r \right)^{-1} \Phi^{L\top}_{\mathbf{x}'} \tag{15}$$

$$= \Phi^r_{\mathbf{x}'} L \left( \frac{1}{\sigma^2} L^\top \Phi^{r\top}_{\mathbf{x}} \Phi^r_{\mathbf{x}} L + I_r \right)^{-1} L^\top \Phi^{r\top}_{\mathbf{x}'}. \tag{16}$$

*Proof of Theorem 3.2.* Consider the Cholesky decomposition $LL^\top = B^\top B$ with lower triangular $L \in \mathbb{R}^{r \times r}$. Using the features $\phi^L(x) = L^\top \phi^r(x)$ gives the kernel

$$k^L(x, x') := \phi^{r\top}(x) L L^\top \phi^r(x') = \phi^{r\top}(x) B^\top B \phi^r(x') = k^B(x, x'). \tag{34}$$

Thus, using the features $\phi^L$ is equivalent to using the features $\phi^B$. Further, define

$$\tilde{B} := BL^{-\top} \in \mathbb{R}^{(m+r) \times r}, \quad \tilde{\phi}^r(x) := L^\top \phi^r(x) \in \mathbb{R}^r. \tag{35}$$

Then, $\tilde{\phi}^{\tilde{B}}(x) = \tilde{B}\tilde{\phi}^r(x) = B\phi^r(x) = \phi^B(x)$. Applying Theorem 3.1 with $\tilde{B}$ and $\tilde{\phi}^r$ in the second equality, and using $\tilde{B}^\top \tilde{B} = I_r$ in the third equality, we get

$$S_{\mathbf{x}',\mathbf{x}'}^{B} = S_{\mathbf{x}',\mathbf{x}'}^{\tilde{B}} \tag{36}$$

$$= \tilde{\Phi}_{\mathbf{x}'}^r \left( \frac{1}{\sigma^2} \tilde{\Phi}_{\mathbf{x}}^{r\top} \tilde{\Phi}_{\mathbf{x}}^r + (\tilde{B}^\top \tilde{B})^{-1} \right)^{-1} \tilde{\Phi}_{\mathbf{x}'}^{r\top} \tag{37}$$

$$= \Phi_{\mathbf{x}'}^r L \left( \frac{1}{\sigma^2} L^\top \Phi_{\mathbf{x}}^{r\top} \Phi_{\mathbf{x}}^r L + I_r \right)^{-1} L^\top \Phi_{\mathbf{x}'}^{r\top}. \tag{38}$$

$\square$

**Theorem 3.3.** *Using the approximation of NTK features via $x \mapsto \phi^B(x) = B\phi^r(x)$ (or, equivalently, $\phi^L(x)$) always gives higher predictive uncertainty than using the BLL features $x \mapsto \phi^r(x)$. In other words,*

$$S_{\mathbf{x}',\mathbf{x}'}^{B} \succeq S_{\mathbf{x}',\mathbf{x}'}^{\mathrm{bll}}. \tag{17}$$

*Proof of Theorem 3.3.* We have $B^\top B = A^\top A + I_r \succeq I_r$. Thus, using the first part of Theorem 3.1:

$$S_{\mathbf{x}',\mathbf{x}'}^{B} = \Phi_{\mathbf{x}'}^r \left( \frac{1}{\sigma^2} \Phi_{\mathbf{x}}^{r\top} \Phi_{\mathbf{x}}^r + (B^\top B)^{-1} \right)^{-1} \Phi_{\mathbf{x}'}^{r\top} \tag{39}$$

$$\succeq \Phi_{\mathbf{x}'}^r \left( \frac{1}{\sigma^2} \Phi_{\mathbf{x}}^{r\top} \Phi_{\mathbf{x}}^r + I_r \right)^{-1} \Phi_{\mathbf{x}'}^{r\top} \tag{40}$$

$$= S_{\mathbf{x}',\mathbf{x}'}^{\mathrm{bll}}. \tag{41}$$

Thus the predictive covariance of our method dominates the predictive covariance of Bayesian Last Layer. $\square$

## A.2. Matrix Concentration Bound

In our analysis we will rely on the following matrix concentration bound:

**Lemma A.1** (Matrix Bernstein inequality)**.** *Let $Z_1, \ldots, Z_n \in \mathbb{R}^{m \times r}$ be independent random matrices with $\mathbb{E}[Z_i] = 0$ and $\|Z_i\|_2 \leq R$ almost surely. Then for $n \geq \frac{8}{3} \log\left( \frac{m+r}{\delta} \right)$: With probability $1 - \delta$,*

$$\left\| \frac{1}{n} \sum_{i=1}^n Z_i \right\|_2 \leq R \sqrt{\frac{8}{3n} \log\left( \frac{m+r}{\delta} \right)}. \tag{42}$$

*Proof.* Define $\sigma^2 := \max \left\{ \left\| \sum_{i=1}^n \mathbb{E}[Z_i Z_i^\top] \right\|_2, \left\| \sum_{i=1}^n \mathbb{E}[Z_i^\top Z_i] \right\|_2 \right\}$. By Tropp (2012), Theorem 1.6: For all $t \geq 0$,

$$\mathbb{P}\left( \left\| \sum_{i=1}^n Z_i \right\|_2 \geq t \right) \leq (m+r) \exp\left( \frac{-t^2/2}{\sigma^2 + Rt/3} \right). \tag{43}$$

Applying this with $t = n\tau$ gives

$$\mathbb{P}\left( \left\| \sum_{i=1}^n Z_i \right\|_2 \geq n\tau \right) \leq (m+r) \exp\left( \frac{-n^2\tau^2/2}{\sigma^2 + Rn\tau/3} \right). \tag{44}$$

Due to $\|Z_i Z_i^\top\|_2 = \|Z_i^\top Z_i\|_2 = \|Z_i\|_2^2 \leq R^2$ we have $\sigma^2 \leq nR^2$. Thus for $\tau \leq R$,

$$\mathbb{P}\left( \left\| \frac{1}{n} \sum_{i=1}^n Z_i \right\|_2 \geq \tau \right) \leq (m+r) \exp\left( \frac{-3n\tau^2}{8R^2} \right). \tag{45}$$

Now let $\delta > 0$. Then $(m+r) \exp\left( \frac{-3n\tau^2}{8R^2} \right) = \delta$ is equivalent to $\tau = R \sqrt{\frac{8}{3n} \log\left( \frac{m+r}{\delta} \right)}$. For $n \geq \frac{8}{3} \log\left( \frac{m+r}{\delta} \right)$ we have $\tau = R \sqrt{\frac{8}{3n} \log\left( \frac{m+r}{\delta} \right)} \leq R$, and can thus apply inequality (45) to get our result. $\square$

## A.3. Proof of Theorem 3.4

**Theorem 3.4.** *Define the true map $A \in \mathbb{R}^{m \times r}$ by*

$$\min_A \mathbb{E}_{x \sim P_X} \left[ \|\phi^m(x) - A\phi^r(x)\|_2^2 \right]. \tag{18}$$

*Consider its approximation $\hat{A}$ from $N$ data points. Assume that the train points are in a compact set and thus there is $K$ such that for all $x$, $\|\phi^m(x)\|_2 \leq K$ and $\|\phi^r(x)\|_2 \leq K$. Then, there is a constant $K'$ depending on $K$ and $\lambda_{\min}(\mathbb{E}_{x \sim P_X}[\phi^r(x)\phi^r(x)^\top]) > 0$, such that: For any $\delta > 0$, there is $N$ large enough such that with probability $1 - \delta$ over iid samples $\mathbf{x}_1, \ldots, \mathbf{x}_N \sim P_X$,*

$$\|\hat{A} - A\|_2 \leq K' \sqrt{\frac{\log\left(2(m+r)/\delta\right)}{N}}. \tag{19}$$

Before starting the proof, note that if $P_X$ has compact support (i.e. the inputs $x$ come from a compact set), there must exist a constant $K$ such that $\|\phi^r(x)\|_2 \leq K$ and $\|\phi^m(x)\|_2 \leq K$ for almost all $x \sim P_X$. This is due to continuity of the feature maps. Further, recall that $(\phi^m(x), \phi^r(x))$ is equal to the parameter-gradient of the neural network backbone at the MAP estimate $\hat{\theta}$. Under NTK-parameterization or $\mu$-parameterization, the parameter-gradient will stay $O(1)$, and in particular it should not scale with $m + r$ (Jacot et al., 2018; Yang & Hu, 2020).

*Proof of Theorem 3.4.* For notational brevity, define the random variables $\phi^m := \phi^m(x) \in \mathbb{R}^m$, $\phi^r := \phi^r(x) \in \mathbb{R}^r$ for $x \sim P_X$, and similarly $\phi_i^m := \phi^m(\mathbf{x}_i)$, $\phi_i^r := \phi^r(\mathbf{x}_i)$ for $\mathbf{x}_i \sim P_X$. The true map is given by

$$A = \mathbb{E}\left[\phi^m \phi^{r\top}\right] \mathbb{E}\left[\phi^r \phi^{r\top}\right]^{-1} =: C_{mr} \Sigma_{rr}^{-1} \in \mathbb{R}^{m \times r}, \tag{46}$$

while its empirical estimate is

$$\hat{A} = \left(\frac{1}{N} \sum_{i=1}^N \phi_i^m \phi_i^{r\top}\right) \left(\frac{1}{N} \sum_{i=1}^N \phi_i^r \phi_i^{r\top}\right)^{-1} =: \hat{C}_{mr} \hat{\Sigma}_{rr}^{-1} \in \mathbb{R}^{m \times r}. \tag{47}$$

Then,

$$\|\hat{A} - A\|_2 = \|\hat{C}_{mr} \hat{\Sigma}_{rr}^{-1} - C_{mr} \Sigma_{rr}^{-1}\|_2 \tag{48}$$

$$\leq \left\|\hat{C}_{mr} - C_{mr}\right\|_2 \left\|\hat{\Sigma}_{rr}^{-1}\right\|_2 + \|C_{mr}\|_2 \left\|\hat{\Sigma}_{rr}^{-1} - \Sigma_{rr}^{-1}\right\|_2. \tag{49}$$

First, note that

$$\|C_{mr}\|_2 = \left\|\mathbb{E}[\phi^m \phi^{r\top}]\right\|_2 \leq \mathbb{E}\left[\left\|\phi^m \phi^{r\top}\right\|_2\right] \leq K^2. \tag{50}$$

For the first term, consider the independent random matrices $\phi_i^m \phi_i^{r\top} - C_{mr} \in \mathbb{R}^{m \times r}$. Then $\mathbb{E}[\phi_i^m \phi_i^{r\top} - C_{mr}] = 0$ and

$$\|\phi_i^m \phi_i^{r\top} - C_{mr}\|_2 \leq \|\phi_i^m \phi_i^{r\top}\|_2 + \|C_{mr}\|_2 \leq 2K^2. \tag{51}$$

Applying Lemma A.1 gives with probability $1 - \delta/2$, for $N \geq \frac{8}{3} \log\left(\frac{m+r}{\delta/2}\right)$:

$$\left\|\hat{C}_{mr} - C_{mr}\right\|_2 \leq 2K^2 \sqrt{\frac{8}{3N} \log\left(\frac{m+r}{\delta/2}\right)}. \tag{52}$$

For the second term, consider the independent random matrices $\phi_i^r \phi_i^{r\top} - \Sigma_{rr} \in \mathbb{R}^{r \times r}$. Then $\mathbb{E}[\phi_i^r \phi_i^{r\top} - \Sigma_{rr}] = 0$ and

$$\|\phi_i^r \phi_i^{r\top} - \Sigma_{rr}\|_2 \leq \|\phi_i^r \phi_i^{r\top}\|_2 + \|\mathbb{E}[\phi^r \phi^{r\top}]\|_2 \leq K^2 + \mathbb{E}[\|\phi^r \phi^{r\top}\|_2] \leq 2K^2. \tag{53}$$

Applying Lemma A.1 gives with probability $1 - \delta/2$, for $N \geq \frac{8}{3} \log\left(\frac{2r}{\delta/2}\right)$:

$$\left\|\hat{\Sigma}_{rr} - \Sigma_{rr}\right\|_2 \leq 2K^2 \sqrt{\frac{8}{3N} \log\left(\frac{2r}{\delta/2}\right)}. \tag{54}$$

Further, note that

$$\|\Sigma_{rr}^{-1}\|_2 = \frac{1}{\lambda_{\min}(\Sigma_{rr})}. \tag{55}$$

We have $2K^2\sqrt{\frac{8}{3N}\log\left(\frac{2r}{\delta/2}\right)} \leq \frac{1}{2}\lambda_{\min}(\Sigma_{rr})$ if and only if $N \geq \left(\frac{4K^2}{\lambda_{\min}(\Sigma_{rr})}\right)^2 \frac{8}{3}\log\left(\frac{2r}{\delta/2}\right)$. In this case, using Weyl's inequality,

$$|\lambda_{\min}(\hat{\Sigma}_{rr}) - \lambda_{\min}(\Sigma_{rr})| \leq \|\hat{\Sigma}_{rr} - \Sigma_{rr}\|_2 \leq \frac{1}{2}\lambda_{\min}(\Sigma_{rr}). \tag{56}$$

This implies $\lambda_{\min}(\hat{\Sigma}_{rr}) \geq \frac{1}{2}\lambda_{\min}(\Sigma_{rr})$ and thus

$$\|\hat{\Sigma}_{rr}^{-1}\|_2 \leq \frac{2}{\lambda_{\min}(\Sigma_{rr})}. \tag{57}$$

Finally, using the resolvent identity $A^{-1} - B^{-1} = A^{-1}(B - A)B^{-1}$ gives

$$\|\hat{\Sigma}_{rr}^{-1} - \Sigma_{rr}^{-1}\|_2 \leq \|\hat{\Sigma}_{rr}^{-1}\|_2\|\Sigma_{rr} - \hat{\Sigma}_{rr}\|_2\|\Sigma_{rr}^{-1}\|_2 \tag{58}$$

$$\leq \frac{2}{\lambda_{\min}(\Sigma_{rr})^2}2K^2\sqrt{\frac{8}{3N}\log\left(\frac{2r}{\delta/2}\right)}. \tag{59}$$

Summing up gives (for $m \geq r$)

$$\|\hat{A} - A\|_2 \leq \frac{2}{\lambda_{\min}(\Sigma_{rr})}2K^2\sqrt{\frac{8}{3N}\log\left(\frac{m+r}{\delta/2}\right)} + K^2\frac{2}{\lambda_{\min}(\Sigma_{rr})^2}2K^2\sqrt{\frac{8}{3N}\log\left(\frac{2r}{\delta/2}\right)} \tag{60}$$

$$\leq \left(1 + \frac{K^2}{\lambda_{\min}(\Sigma_{rr})}\right)\frac{4K^2}{\lambda_{\min}(\Sigma_{rr})}\sqrt{\frac{8}{3N}\log\left(\frac{m+r}{\delta/2}\right)}. \tag{61}$$

The result follows with $K' := \sqrt{8/3}\left(1 + \frac{K^2}{\lambda_{\min}(\Sigma_{rr})}\right)\frac{4K^2}{\lambda_{\min}(\Sigma_{rr})}$.

$\square$

### A.4. Proof of Theorem 3.5

**Lemma A.2.** *Consider the setting of Theorem 3.5. Define $\eta_k(\delta) := 2K^2\sqrt{\frac{8}{3k}\log(4r/\delta)}$. Then, with probability $1 - \delta$:*

$$\left\|\frac{1}{k}\sum_{i=1}^{k}\left(\phi_{s_i}\phi_{s_i}^\top - \Sigma\right)\right\|_2 \leq \eta_k(\delta), \quad \left\|\frac{1}{N}\sum_{i=1}^{N}\left(\phi_i\phi_i^\top - \Sigma\right)\right\|_2 \leq \eta_N(\delta) \leq \eta_k(\delta). \tag{62}$$

*Proof of Lemma A.2.* For notational brevity, define the random variables $\phi := \phi^L(x) \in \mathbb{R}^r$ for $x \sim P_X$, and similarly $\phi_i := \phi^L(\mathbf{x}_i)$ for $\mathbf{x}_i \sim P_X$. The empirical population matrix is $\sum_{i=1}^{N}\phi_i\phi_i^\top =: N\hat{\Sigma}_N$, and its subsampled estimator by $\frac{N}{k}\sum_{i=1}^{k}\phi_{s_i}\phi_{s_i}^\top =: N\hat{\Sigma}_k$. Recall that $\Sigma := \mathbb{E}[\phi\phi^\top] \in \mathbb{R}^{r\times r}$. We can bound

$$\|\phi_i\phi_i^\top - \Sigma\|_2 \leq \|\phi_i\phi_i^\top\|_2 + \|\mathbb{E}[\phi\phi^\top]\|_2 \leq K^2 + \mathbb{E}\left[\|\phi_i\phi_i^\top\|_2\right] \leq 2K^2. \tag{63}$$

Define $\eta_k(\delta) := 2K^2\sqrt{\frac{8}{3k}\log(4r/\delta)}$. Apply Lemma A.1 with $Z_i = \phi_i\phi_i^\top - \Sigma$ and $\delta/2$ for $Z_{s_1}, \ldots, Z_{s_k}$ and $Z_1, \ldots, Z_N$. Then, we get for $N \geq k \geq \frac{8}{3}\log(4r/\delta)$, with probability $1 - \delta$:

$$\left\|\frac{1}{k}\sum_{i=1}^{k}\left(\phi_{s_i}\phi_{s_i}^\top - \Sigma\right)\right\|_2 \leq \eta_k(\delta), \quad \left\|\frac{1}{N}\sum_{i=1}^{N}\left(\phi_i\phi_i^\top - \Sigma\right)\right\|_2 \leq \eta_N(\delta) \leq \eta_k(\delta). \tag{64}$$

$\square$

**Lemma A.3.** *For all $t \geq 0$, $\left| \frac{1}{1+t} - \frac{1}{1+(1+\epsilon)t} \right| \leq \frac{\epsilon}{4}$. For $\epsilon \leq \frac{1}{2}$, [1] we also have $\left| \frac{1}{1+t} - \frac{1}{1+(1-\epsilon)t} \right| \leq \frac{\epsilon}{2}$.*

*Proof.* For any $c > 0$, consider the function

$$t \mapsto g_c(t) := \left| \frac{1}{1+t} - \frac{1}{1+ct} \right| = |c-1| \frac{t}{(1+t)(1+ct)}. \tag{65}$$

This has derivative

$$\frac{d}{dt} g_c(t) = |c-1| \frac{1 + (c+1)t + ct^2 - t((c+1) + 2ct)}{((1+t)(1+ct))^2} = |c-1| \frac{1 - ct^2}{((1+t)(1+ct))^2}. \tag{66}$$

Thus, it is maximized at $t^* = \frac{1}{\sqrt{c}}$, with maximum value $\frac{|c-1|}{(1+\sqrt{c})^2}$.

Applying this with $c = 1 + \epsilon$,

$$\left| \frac{1}{1+t} - \frac{1}{1+(1+\epsilon)t} \right| \leq \epsilon \frac{1}{(1+\sqrt{1+\epsilon})^2} \leq \frac{\epsilon}{4}. \tag{67}$$

Further, if $\epsilon$ satisfies $1 + \sqrt{1-\epsilon} \geq \sqrt{2}$ (e.g. for $\epsilon \leq \frac{3}{4}$), we get with $c = 1 - \epsilon$,

$$\left| \frac{1}{1+t} - \frac{1}{1+(1-\epsilon)t} \right| \leq \epsilon \frac{1}{(1+\sqrt{1-\epsilon})^2} \leq \frac{\epsilon}{2}. \tag{68}$$

$\square$

**Theorem 3.5.** *Assume that the training and test points are in a compact set, and the feature vector $\phi^L(x) = L^\top \phi^r(x) \in \mathbb{R}^r$ is bounded: For all $x$, $\|\phi^L(x)\|_2 \leq K$. Assume that the smallest eigenvalue of $\Sigma := \mathbb{E}_{x \sim P_X} \left[ \phi^L(x) \phi^L(x)^\top \right] \in \mathbb{R}^{r \times r}$ is positive. Let $N \geq k \geq \frac{8}{3} \log\left(4r/\delta\right) \left( \frac{8K^2}{\lambda_{\min}(\Sigma)} \right)^2$. Then with probability of at least $1 - \delta$ over iid samples $\mathbf{x}_1, \ldots, \mathbf{x}_N \sim P_X$: For any $N'$ test points $\mathbf{x}'_1, \ldots, \mathbf{x}'_{N'}$,*

$$\|S_{\mathbf{x}',\mathbf{x}'}^{B,k} - S_{\mathbf{x}',\mathbf{x}'}^B\|_2 \leq N' \frac{2K^4}{\lambda_{\min}(\Sigma)} \sqrt{\frac{8 \log(4r/\delta)}{3k}}. \tag{23}$$

*Proof of Theorem 3.5.* Lemma A.2 states that for $N \geq k \geq \frac{8}{3} \log\left(\frac{4r}{\delta}\right)$, with probability $1 - \delta$:

$$\Sigma - \eta_k I_r \preceq \hat{\Sigma}_k, \hat{\Sigma}_N \preceq \Sigma + \eta_k I_r, \tag{69}$$

where $\eta_k(\delta) = 2K^2 \sqrt{\frac{8}{3k} \log(4r/\delta)}$. Let $\epsilon_k := \frac{\eta_k(\delta)}{\lambda_{\min}(\Sigma)}$. Due to $\frac{1}{\lambda_{\min}(\Sigma)} \Sigma \succeq I_r$, we get

$$(1 - \epsilon_k)\Sigma \preceq \hat{\Sigma}_k, \hat{\Sigma}_N \preceq (1 + \epsilon_k)\Sigma. \tag{70}$$

We can now further follow

$$(1 - 2\epsilon_k)\hat{\Sigma}_N \preceq \hat{\Sigma}_k \preceq (1 + 4\epsilon_k)\hat{\Sigma}_N. \tag{71}$$

Multiply both sides with $N$. It follows that

$$I_r + \sigma^{-2}(1 - 2\epsilon_k)N\hat{\Sigma}_N \preceq I_r + \sigma^{-2}N\hat{\Sigma}_k \preceq I_r + \sigma^{-2}(1 + 4\epsilon_k)N\hat{\Sigma}_N, \tag{72}$$

and further

$$\left( I_r + \sigma^{-2}(1 + 4\epsilon_k)N\hat{\Sigma}_N \right)^{-1} \preceq \left( I_r + \sigma^{-2}N\hat{\Sigma}_k \right)^{-1} \preceq \left( I_r + \sigma^{-2}(1 - 2\epsilon_k)N\hat{\Sigma}_N \right)^{-1}. \tag{73}$$

---

[1] It suffices that $\epsilon \leq 1 - (\sqrt{2} - 1)^2$. This is given for $\epsilon \leq \frac{1}{2}$, as $(\sqrt{2} - 1)^2 \leq \frac{1}{2}$ is equivalent to $\sqrt{2}(\sqrt{2} - 1) \leq 1$, which is equivalent to $1 \leq \sqrt{2}$.

Substracting $\left(I_r + \alpha N \hat{\Sigma}_N\right)^{-1}$ on all sides gives

$$\left\| \left(I_r + \sigma^{-2} N \hat{\Sigma}_k\right)^{-1} - \left(I_r + \sigma^{-2} N \hat{\Sigma}_N\right)^{-1} \right\|_2 \tag{74}$$

$$\leq \max \left\{ \left\| \left(I_r + \sigma^{-2}(1 + 4\epsilon_k) N \hat{\Sigma}_N\right)^{-1} - \left(I_r + \sigma^{-2} N \hat{\Sigma}_N\right)^{-1} \right\|_2, \left\| \left(I_r + \sigma^{-2}(1 - 2\epsilon_k) N \hat{\Sigma}_N\right)^{-1} - \left(I_r + \sigma^{-2} N \hat{\Sigma}_N\right)^{-1} \right\|_2 \right\} \tag{75}$$

$$= \max \left\{ \max_i \left| \frac{1}{1 + \sigma^{-2}\lambda_i} - \frac{1}{1 + \sigma^{-2}(1 + 4\epsilon_k)\lambda_i} \right|, \max_i \left| \frac{1}{1 + \sigma^{-2}\lambda_i} - \frac{1}{1 + \sigma^{-2}(1 - 2\epsilon_k)\lambda_i} \right| \right\}. \tag{76}$$

We denoted the eigenvalues of $N\hat{\Sigma}_N$ by $\lambda_i \geq 0$. Now let $k$ large enough such that $2\epsilon_k = \frac{2\eta_k(\delta)}{\lambda_{\min}(\Sigma)} \leq \frac{1}{2}$. This is given if $k \geq \frac{8}{3} \log\left(\frac{4r}{\delta}\right) \left(\frac{8K^2}{\lambda_{\min}(\Sigma)}\right)^2$. Then, we can apply Lemma A.3 with $t = \sigma^{-2}\lambda_i$ to get

$$\left\| \left(I_r + \sigma^{-2} N \hat{\Sigma}_k\right)^{-1} - \left(I_r + \sigma^{-2} N \hat{\Sigma}_N\right)^{-1} \right\|_2 \leq \epsilon_k = \frac{\eta_k(\delta)}{\lambda_{\min}(\Sigma)}. \tag{77}$$

Recall that $\eta_k(\delta) = 2K^2 \sqrt{\frac{8}{3k} \log(4r/\delta)}$.

Finally, consider $N'$ test points $\mathbf{x}'_1, \ldots, \mathbf{x}'_{N'}$. $\|\phi^L(\mathbf{x}'_i)\|_2 \leq K$ implies $\|\Phi^L_{\mathbf{x}'}\|_2 \leq \sqrt{N'}K$. Thus,

$$\|S^{B,k}_{\mathbf{x}',\mathbf{x}'} - S^B_{\mathbf{x}',\mathbf{x}'}\|_2 = \sigma^2 \left\| \Phi^L_{\mathbf{x}'} \left( \left(\sigma^2 I_r + N\hat{\Sigma}_N\right)^{-1} - \left(\sigma^2 I_r + N\hat{\Sigma}_k\right)^{-1} \right) \Phi^{L\top}_{\mathbf{x}'} \right\|_2 \tag{78}$$

$$\leq \sigma^2 N' K^2 \frac{1}{\sigma^2} \eta_k(\delta) \frac{1}{\lambda_{\min}(\Sigma)} \tag{79}$$

$$= N' \frac{2K^4}{\lambda_{\min}(\Sigma)} \sqrt{\frac{8}{3k} \log(4r/\delta)}. \tag{80}$$

$\square$

### A.5. Proof of Alternative Bound in Theorem 3.5

In the following, we prove an alternative bound that may be sharper in certain cases.

**Theorem A.4.** *Consider the assumptions of Theorem 3.5. Let $N \geq k \geq \frac{8}{3} \log\left(\frac{4r}{\delta}\right) \left(\frac{4K^2}{\lambda_{\min}(\Sigma)}\right)^2$. Then with probability of at least $1 - \delta$ over iid samples $\mathbf{x}_1, \ldots, \mathbf{x}_N \sim P_X$, for any $N'$ test points $\mathbf{x}'_1, \ldots, \mathbf{x}'_{N'}$,*

$$\|S^{B,k}_{\mathbf{x}',\mathbf{x}'} - S^B_{\mathbf{x}',\mathbf{x}'}\|_2 \leq N' \frac{4K^4}{(\sigma/N + \sigma^{-1}\lambda_{\min}(\Sigma)/2)^2} \frac{1}{N} \sqrt{\frac{8}{3k} \log(4r/\delta)}. \tag{81}$$

*Proof of Theorem A.4.* Applying Lemma A.2 gives for $N \geq k \geq \frac{8}{3} \log\left(\frac{4r}{\delta}\right)$, with probability $1 - \delta$:

$$\|\hat{\Sigma}_k - \hat{\Sigma}_N\|_2 \leq \|\hat{\Sigma}_k - \Sigma\|_2 + \|\Sigma - \hat{\Sigma}_N\|_2 \leq 2\eta_k(\delta). \tag{82}$$

Using the resolvent identity $A^{-1} - B^{-1} = A^{-1}(B - A)B^{-1}$ we get for $k \geq \frac{8}{3} \log\left(\frac{4r}{\delta}\right)$, with probability $1 - \delta$:

$$\left\| \left(I_r + \sigma^{-2} N \hat{\Sigma}_N\right)^{-1} - \left(I_r + \sigma^{-2} N \hat{\Sigma}_k\right)^{-1} \right\|_2 \leq \left\| \left(I_r + \sigma^{-2} N \hat{\Sigma}_N\right)^{-1} \right\|_2 \sigma^{-2} \left\| N\hat{\Sigma}_k - N\hat{\Sigma}_N \right\|_2 \left\| \left(I_r + \sigma^{-2} N \hat{\Sigma}_k\right)^{-1} \right\|_2 \tag{83}$$

$$\leq \frac{1}{1 + \sigma^{-2}N\lambda_{\min}(\hat{\Sigma}_N)} \sigma^{-2} N 2\eta_k(\delta) \frac{1}{1 + \sigma^{-2}N\lambda_{\min}(\hat{\Sigma}_k)} \tag{84}$$

$$= 2\eta_k(\delta) \frac{1}{N} \frac{1}{\sigma/N + \sigma^{-1}\lambda_{\min}(\hat{\Sigma}_N)} \frac{1}{\sigma/N + \sigma^{-1}\lambda_{\min}(\hat{\Sigma}_k)} \tag{85}$$

$$\leq 2\eta_k(\delta) \frac{1}{N} \frac{1}{(\sigma/N + \sigma^{-1}\lambda_{\min}(\Sigma)/2)^2}. \tag{86}$$

Recall that $\eta_k(\delta) = 2K^2\sqrt{\frac{8}{3k}\log(4r/\delta)}$. In the last step, we used $\lambda_{\min}(\hat{\Sigma}_N) \geq \lambda_{\min}(\Sigma) - \eta_k(\delta)$ and $\lambda_{\min}(\hat{\Sigma}_k) \geq \lambda_{\min}(\Sigma) - \eta_k(\delta)$, and further chose $k$ large enough such that $\eta_k(\delta) \leq \frac{1}{2}\lambda_{\min}(\Sigma)$, i.e. $k \geq \frac{8}{3}\log\left(\frac{4r}{\delta}\right)\left(\frac{4K^2}{\lambda_{\min}(\Sigma)}\right)^2$.

The bound for $\|S^{B,k}_{\mathbf{x}',\mathbf{x}'} - S^B_{\mathbf{x}',\mathbf{x}'}\|_2$ follows in the same way as in the original proof. $\qquad\square$

## B. On the Exactness of the Linear Projection under Low-Rank eNTK

**Lemma B.1.** *Let $x_1, \ldots, x_N \in \mathbb{R}^d$ be input points, and let $\Phi^p_{\mathbf{x}} = [\Phi^m_{\mathbf{x}}, \Phi^r_{\mathbf{x}}] \in \mathbb{R}^{N\times(m+r)}$ denote the empirical NTK feature matrix. Assume that $N \geq r$, that the last-layer feature matrix has full column rank, $\mathrm{rank}(\Phi^r_{\mathbf{x}}) = r$, and that the eNTK matrix $k^p_{\mathbf{x},\mathbf{x}} = \Phi^p_{\mathbf{x}}\Phi^{p\top}_{\mathbf{x}}$ has rank at most $r$. Then, there exists a matrix $A_* \in \mathbb{R}^{m\times r}$ such that $\Phi^m_{\mathbf{x}} = \Phi^r_{\mathbf{x}} A^\top_*$. Equivalently, if $B_* := \begin{pmatrix} A_* \\ I_r \end{pmatrix} \in \mathbb{R}^{(m+r)\times r}$, then*

$$\Phi^p_{\mathbf{x}} = \Phi^r_{\mathbf{x}} B^\top_*.$$

*Moreover, the least-squares solution is unique*

$$A_* = \Phi^{m\top}_{\mathbf{x}}\Phi^r_{\mathbf{x}}(\Phi^{r\top}_{\mathbf{x}}\Phi^r_{\mathbf{x}})^{-1}.$$

*Proof.* Since $k^p_{\mathbf{x},\mathbf{x}} = \Phi^p_{\mathbf{x}}\Phi^{p\top}_{\mathbf{x}}$, we have $\mathrm{rank}(k^p_{\mathbf{x},\mathbf{x}}) = \mathrm{rank}(\Phi^p_{\mathbf{x}})$. By assumption, $\mathrm{rank}(\Phi^p_{\mathbf{x}}) \leq r$. On the other hand, $\Phi^r_{\mathbf{x}}$ is a block of $\Phi^p_{\mathbf{x}}$, so every column of $\Phi^r_{\mathbf{x}}$ is also a column of $\Phi^p_{\mathbf{x}}$. Hence $\mathrm{col}(\Phi^r_{\mathbf{x}}) \subseteq \mathrm{col}(\Phi^p_{\mathbf{x}})$, which implies $\mathrm{rank}(\Phi^p_{\mathbf{x}}) \geq \mathrm{rank}(\Phi^r_{\mathbf{x}})$. Therefore $\mathrm{rank}(\Phi^p_{\mathbf{x}}) = r$.

We now have $\mathrm{col}(\Phi^r_{\mathbf{x}}) \subseteq \mathrm{col}(\Phi^p_{\mathbf{x}})$ and both spaces have the same dimension $r$. It follows that $\mathrm{col}(\Phi^r_{\mathbf{x}}) = \mathrm{col}(\Phi^p_{\mathbf{x}})$. In particular, each column of $\Phi^m_{\mathbf{x}}$ belongs to $\mathrm{col}(\Phi^r_{\mathbf{x}})$. Therefore there exists a matrix $A^\top_* \in \mathbb{R}^{r\times m}$ such that $\Phi^m_{\mathbf{x}} = \Phi^r_{\mathbf{x}} A^\top_*$. This proves the first claim.

Defining $B_* := \begin{pmatrix} A_* \\ I_r \end{pmatrix}$, we obtain

$$\Phi^r_{\mathbf{x}} B^\top_* = [\Phi^r_{\mathbf{x}} A^\top_*, \Phi^r_{\mathbf{x}}] = [\Phi^m_{\mathbf{x}}, \Phi^r_{\mathbf{x}}] = \Phi^p_{\mathbf{x}},$$

which proves the equivalent factorization.

Finally, since $\mathrm{rank}(\Phi^r_{\mathbf{x}}) = r$, the matrix $\Phi^{r\top}_{\mathbf{x}}\Phi^r_{\mathbf{x}}$ is invertible. The equation $\Phi^m_{\mathbf{x}} = \Phi^r_{\mathbf{x}} A^\top_*$ therefore has the unique least-squares solution

$$A^\top_* = (\Phi^{r\top}_{\mathbf{x}}\Phi^r_{\mathbf{x}})^{-1}\Phi^{r\top}_{\mathbf{x}}\Phi^m_{\mathbf{x}}.$$

Transposing both sides yields

$$A_* = \Phi^{m\top}_{\mathbf{x}}\Phi^r_{\mathbf{x}}(\Phi^{r\top}_{\mathbf{x}}\Phi^r_{\mathbf{x}})^{-1}.$$

This completes the proof. $\qquad\square$

The previous lemma gives an exact recovery result: if the empirical NTK feature matrix has rank at most $r$ and the last-layer features span an $r$-dimensional subspace, then the non-last-layer features are exactly representable as a linear combination of the last-layer features. In this regime, the proposed projection introduces no approximation error on the data. This motivates checking whether the empirical NTK has low effective rank in the models used in our experiments.

Figures 3 and 4 report the spectrum of the empirical NTK Gram matrix for the MLP used in the UCI regression experiments and the CNN used in the CIFAR-10 experiments, respectively. In both cases, the spectrum exhibits strong concentration, i.e., a relatively small number of leading eigenvalues accounts for most of the trace. This supports the premise that, in these models, the empirical NTK contains a dominant low-dimensional structure.

However, spectral decay alone does not imply that this low-dimensional structure is fully captured by the span of the last-layer features. The exact recovery lemma above gives a sufficient condition for this to happen, but in practice the empirical NTK need not be exactly rank $r$, and the last-layer span may only approximately capture the relevant directions. We therefore introduce a relative quasi-low-rank residual that directly measures the part of the full eNTK feature matrix that lies outside the span of the last-layer features. The next definition formalizes this approximate setting.

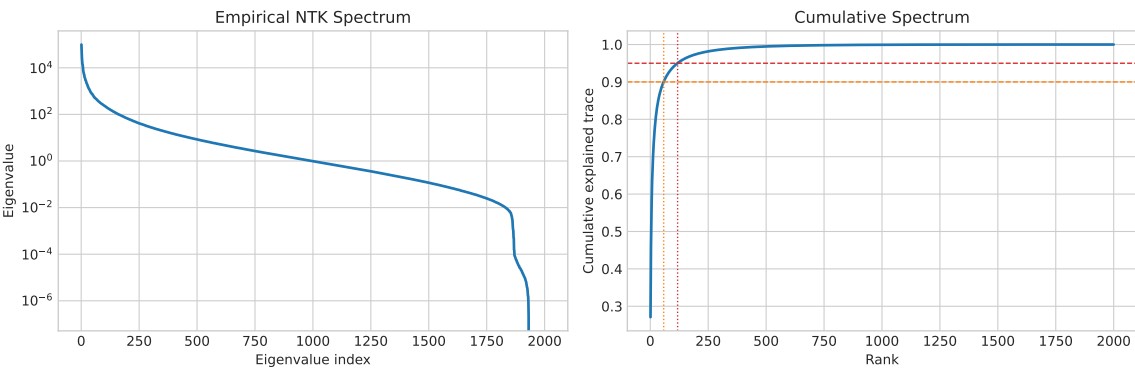

*Figure 3.* Empirical NTK spectrum for the MLP used in the UCI regression experiments. **Left:** eigenvalues of the empirical NTK Gram matrix in decreasing order. **Right:** cumulative fraction of the NTK trace explained by the leading eigenvalues. The fast spectral decay indicates strong concentration of the NTK in a relatively low-dimensional subspace, motivating our low-rank feature approximation.

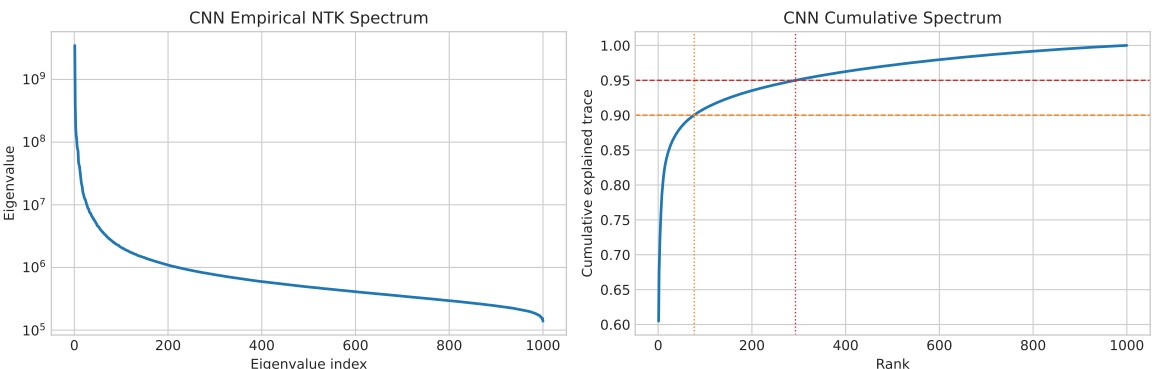

*Figure 4.* Empirical NTK spectrum for the CNN used in the CIFAR-10 classification experiments. **Left:** eigenvalues of the empirical NTK Gram matrix in decreasing order. **Right:** cumulative explained trace as a function of rank. As in the regression setting, most of the trace is captured by a small subset of directions, supporting our low-rank NTK approximation in image classification.

**Definition B.2** (Relative quasi low-rank residual). *Let* $\Phi_{\mathbf{x}}^p = [\Phi_{\mathbf{x}}^m, \Phi_{\mathbf{x}}^r] \in \mathbb{R}^{N \times (m+r)}$ *be the empirical NTK feature matrix, and assume that* $\Phi_{\mathbf{x}}^r$ *has full column rank* $r$. *Define the orthogonal projector onto the column space of* $\Phi_{\mathbf{x}}^r$ *by*

$$P_r := \Phi_{\mathbf{x}}^r (\Phi_{\mathbf{x}}^{r\top} \Phi_{\mathbf{x}}^r)^{-1} \Phi_{\mathbf{x}}^{r\top} \in \mathbb{R}^{N \times N}.$$

*We define the relative quasi low-rank residual by*

$$\varepsilon_{\mathbf{x}} := \|(I_N - P_r)\Phi_{\mathbf{x}}^p\|_2.$$

*Equivalently, since* $(I_N - P_r)\Phi_{\mathbf{x}}^r = 0$*, we have*

$$\varepsilon_{\mathbf{x}} = \|(I_N - P_r)\Phi_{\mathbf{x}}^m\|_2.$$

The quantity $\varepsilon_{\mathbf{x}}$ measures how much of the full eNTK feature matrix lies outside the span of the last-layer features on the training set. When $\varepsilon_{\mathbf{x}} = 0$, we recover the exact setting of the previous lemma. For $\varepsilon_{\mathbf{x}} > 0$, it quantifies the smallest possible feature approximation error attainable by a linear predictor based on the last-layer features.

Since our method uses the least-squares map from $\Phi_{\mathbf{x}}^r$ to $\Phi_{\mathbf{x}}^m$, the relevant question is whether this map achieves the residual above. In the next proposition we show that it does. The feature approximation error induced by the learned linear map is exactly equal to the relative quasi low-rank residual.

**Proposition B.3.** *Let* $\widehat{A} := \Phi_{\mathbf{x}}^{m\top} \Phi_{\mathbf{x}}^r (\Phi_{\mathbf{x}}^{r\top} \Phi_{\mathbf{x}}^r)^{-1} \in \mathbb{R}^{m \times r}$ *be the least-squares estimator, and define* $\Phi_{\mathbf{x}}^B := \Phi_{\mathbf{x}}^r \widehat{B}^\top$ *with* $\widehat{B} := \begin{pmatrix} \widehat{A} \\ I_r \end{pmatrix}$. *Then*

$$\|\Phi_{\mathbf{x}}^p - \Phi_{\mathbf{x}}^B\|_2 = \varepsilon_{\mathbf{x}}.$$

*In particular, the training feature approximation error is exactly equal to the relative quasi low-rank residual.*

*Proof.* By definition of $\widehat{A}$,

$$\widehat{A}^\top = (\Phi_\mathbf{x}^{r\top} \Phi_\mathbf{x}^r)^{-1} \Phi_\mathbf{x}^{r\top} \Phi_\mathbf{x}^m.$$

Hence

$$\Phi_\mathbf{x}^r \widehat{A}^\top = \Phi_\mathbf{x}^r (\Phi_\mathbf{x}^{r\top} \Phi_\mathbf{x}^r)^{-1} \Phi_\mathbf{x}^{r\top} \Phi_\mathbf{x}^m = P_r \Phi_\mathbf{x}^m.$$

Therefore

$$\Phi_\mathbf{x}^m - \Phi_\mathbf{x}^r \widehat{A}^\top = (I_N - P_r) \Phi_\mathbf{x}^m.$$

Since

$$\Phi_\mathbf{x}^p - \Phi_\mathbf{x}^B = [\Phi_\mathbf{x}^m, \Phi_\mathbf{x}^r] - [\Phi_\mathbf{x}^r \widehat{A}^\top, \Phi_\mathbf{x}^r] = [\Phi_\mathbf{x}^m - \Phi_\mathbf{x}^r \widehat{A}^\top, 0],$$

we obtain

$$\|\Phi_\mathbf{x}^p - \Phi_\mathbf{x}^B\|_2 = \|\Phi_\mathbf{x}^m - \Phi_\mathbf{x}^r \widehat{A}^\top\|_2 = \|(I_N - P_r)\Phi_\mathbf{x}^m\|_2 = \varepsilon_\mathbf{x}.$$

This proves the claim. $\qquad\square$

This result shows that the approximation error of the learned map is controlled by a geometric property of the full feature matrix, namely its distance to the span of the last-layer features. To understand the effect on uncertainty estimation, it remains to show how this feature-level error propagates to the posterior covariance. To do so, let $\Phi_{\mathbf{x}'}^B$ denote the approximate test feature matrix induced by the same learned map $\widehat{A}$, and define

$$\varepsilon_{\mathbf{x}'} := \|\Phi_{\mathbf{x}'}^p - \Phi_{\mathbf{x}'}^B\|_2.$$

We now compare the predictive covariance computed from the full eNTK features with that obtained from the projected features. The next result shows that if the train and test feature approximation errors are small, then the resulting predictive covariance error is also small.

**Proposition B.4.** *Let $\Phi_\mathbf{x}^B$ and $\Phi_{\mathbf{x}'}^B$ be induced by the least-squares map $\widehat{A}$ above. Then*

$$\|S_{\mathbf{x}',\mathbf{x}'}^{\mathrm{ntk}} - S_{\mathbf{x}',\mathbf{x}'}^B\|_2 \leq \left(\|\Phi_{\mathbf{x}'}^p\|_2 + \|\Phi_{\mathbf{x}'}^B\|_2\right)\varepsilon_{\mathbf{x}'} + \frac{\|\Phi_{\mathbf{x}'}^p\|_2 \|\Phi_{\mathbf{x}'}^B\|_2}{\sigma^2}\left(\|\Phi_\mathbf{x}^p\|_2 + \|\Phi_\mathbf{x}^B\|_2\right)\varepsilon_\mathbf{x}.$$

*Thus if both the training residual $\varepsilon_\mathbf{x}$ and the test residual $\varepsilon_{\mathbf{x}'}$ are small, the predictive covariance induced by the projected features is close to that of the full eNTK model.*

*Proof.* Define

$$M_p := I_{m+r} + \frac{1}{\sigma^2} \Phi_\mathbf{x}^{p\top} \Phi_\mathbf{x}^p, \qquad M_B := I_{m+r} + \frac{1}{\sigma^2} \Phi_\mathbf{x}^{B\top} \Phi_\mathbf{x}^B.$$

Then

$$S_{\mathbf{x}',\mathbf{x}'}^{\mathrm{ntk}} = \Phi_{\mathbf{x}'}^p M_p^{-1} \Phi_{\mathbf{x}'}^{p\top}, \qquad S_{\mathbf{x}',\mathbf{x}'}^B = \Phi_{\mathbf{x}'}^B M_B^{-1} \Phi_{\mathbf{x}'}^{B\top}.$$

Subtracting and adding intermediate terms gives

$$S_{\mathbf{x}',\mathbf{x}'}^{\mathrm{ntk}} - S_{\mathbf{x}',\mathbf{x}'}^B = (\Phi_{\mathbf{x}'}^p - \Phi_{\mathbf{x}'}^B) M_p^{-1} \Phi_{\mathbf{x}'}^{p\top} + \Phi_{\mathbf{x}'}^B (M_p^{-1} - M_B^{-1}) \Phi_{\mathbf{x}'}^{p\top} + \Phi_{\mathbf{x}'}^B M_B^{-1} (\Phi_{\mathbf{x}'}^{p\top} - \Phi_{\mathbf{x}'}^{B\top}).$$

Taking operator norms and using $M_p \succeq I_{m+r}$ and $M_B \succeq I_{m+r}$, hence $\|M_p^{-1}\|_2 \leq 1$ and $\|M_B^{-1}\|_2 \leq 1$, yields

$$\|S_{\mathbf{x}',\mathbf{x}'}^{\mathrm{ntk}} - S_{\mathbf{x}',\mathbf{x}'}^B\|_2 \leq \left(\|\Phi_{\mathbf{x}'}^p\|_2 + \|\Phi_{\mathbf{x}'}^B\|_2\right)\varepsilon_{\mathbf{x}'} + \|\Phi_{\mathbf{x}'}^p\|_2 \|\Phi_{\mathbf{x}'}^B\|_2 \|M_p^{-1} - M_B^{-1}\|_2.$$

Next we use the identity $A^{-1} - B^{-1} = A^{-1}(B - A)B^{-1}$, valid for any invertible matrices $A$ and $B$. Applying it with $A = M_p$ and $B = M_B$ gives

$$M_p^{-1} - M_B^{-1} = M_p^{-1}(M_B - M_p)M_B^{-1}.$$

Therefore

$$\|M_p^{-1} - M_B^{-1}\|_2 \leq \|M_p^{-1}\|_2 \|M_B - M_p\|_2 \|M_B^{-1}\|_2 \leq \|M_B - M_p\|_2.$$

Moreover,

$$M_B - M_p = \frac{1}{\sigma^2}\left(\Phi_\mathbf{x}^{B\top}\Phi_\mathbf{x}^B - \Phi_\mathbf{x}^{p\top}\Phi_\mathbf{x}^p\right),$$

and

$$\Phi_{\mathbf{x}}^{B\top}\Phi_{\mathbf{x}}^{B} - \Phi_{\mathbf{x}}^{p\top}\Phi_{\mathbf{x}}^{p} = \Phi_{\mathbf{x}}^{B\top}(\Phi_{\mathbf{x}}^{B} - \Phi_{\mathbf{x}}^{p}) + (\Phi_{\mathbf{x}}^{B} - \Phi_{\mathbf{x}}^{p})^{\top}\Phi_{\mathbf{x}}^{p}.$$

Hence

$$\|\Phi_{\mathbf{x}}^{B\top}\Phi_{\mathbf{x}}^{B} - \Phi_{\mathbf{x}}^{p\top}\Phi_{\mathbf{x}}^{p}\|_{2} \leq \left(\|\Phi_{\mathbf{x}}^{p}\|_{2} + \|\Phi_{\mathbf{x}}^{B}\|_{2}\right)\varepsilon_{\mathbf{x}},$$

and thus

$$\|M_{p}^{-1} - M_{B}^{-1}\|_{2} \leq \frac{1}{\sigma^{2}}\left(\|\Phi_{\mathbf{x}}^{p}\|_{2} + \|\Phi_{\mathbf{x}}^{B}\|_{2}\right)\varepsilon_{\mathbf{x}}.$$

Substituting this into the earlier bound proves the result. $\qquad\square$

The previous proposition shows that the error in the predictive covariance is controlled by two quantities: the training residual $\varepsilon_{\mathbf{x}}$, which measures how well the projected features reproduce the training covariance, and the test residual $\varepsilon_{\mathbf{x}'}$, which measures how well the same map generalizes to the test points. The dependence on $1/\sigma^2$ is also natural: when the observation noise is small, the posterior becomes more sensitive to errors in the training covariance.

Combining Lemma B.1 and Proposition B.4, we obtain the following picture. If the full eNTK feature matrix lies exactly in the span of the last-layer features, then the learned linear map recovers the exact feature representation and the posterior covariance is exact. If the full eNTK feature matrix is only approximately contained in that span, then the projection residual $\varepsilon_{\mathbf{x}}$ controls the train-side feature error, and the resulting predictive covariance differs from the full eNTK posterior by an amount controlled by $\varepsilon_{\mathbf{x}}$ and the corresponding test residual $\varepsilon_{\mathbf{x}'}$.

Our spectral experiments support our interpretation. We observe strong eigenvalue decay in the empirical NTK, with fewer than $r$ directions explaining most of the trace. This indicates that the eNTK is effectively low-rank, and therefore suggests that the residuals $\varepsilon_{\mathbf{x}}$ and $\varepsilon_{\mathbf{x}'}$ should be small in practice.

## C. Efficient Computation of the Feature Transform $L$

In this section we show how to efficiently compute the lower triangular $L \in \mathbb{R}^{r\times r}$ that forms the Cholesky decomposition of

$$B^{\top}B = A^{\top}A + I_{r} = \left(\Phi_{\mathbf{x}}^{r}(\Phi_{\mathbf{x}}^{r\top}\Phi_{\mathbf{x}}^{r})^{-1}\right)^{\top}\Phi_{\mathbf{x}}^{m}\Phi_{\mathbf{x}}^{m\top}\left(\Phi_{\mathbf{x}}^{r}(\Phi_{\mathbf{x}}^{r\top}\Phi_{\mathbf{x}}^{r})^{-1}\right) + I_{r}. \tag{87}$$

Remember that $\Phi_{\mathbf{x}}^{r} \in \mathbb{R}^{N\times r}$ and $\Phi_{\mathbf{x}}^{m} \in \mathbb{R}^{N\times m}$. In our application, $m > N > r$.

First, we compute $\Phi_{\mathbf{x}}^{r}(\Phi_{\mathbf{x}}^{r\top}\Phi_{\mathbf{x}}^{r})^{-1} \in \mathbb{R}^{N\times r}$ in $O(Nr^2 + r^3)$ time and $O(Nr + r^2)$ memory in 3 steps: Compute $\Phi_{\mathbf{x}}^{r\top}\Phi_{\mathbf{x}}^{r} \in \mathbb{R}^{r\times r}$. Next, compute the Cholesky decomposition of $\Phi_{\mathbf{x}}^{r\top}\Phi_{\mathbf{x}}^{r} \in \mathbb{R}^{r\times r}$. Finally, compute $\Phi_{\mathbf{x}}^{r}(\Phi_{\mathbf{x}}^{r\top}\Phi_{\mathbf{x}}^{r})^{-1} \in \mathbb{R}^{N\times r}$ by two triangular solves. Subsampling replaces $N$ with $k$.

Alternatively, we could compute $\Phi_{\mathbf{x}}^{r}(\Phi_{\mathbf{x}}^{r\top}\Phi_{\mathbf{x}}^{r})^{-1} \in \mathbb{R}^{N\times r}$ by first forming the QR decomposition $\Phi_{\mathbf{x}}^{r} = QR$, and then using $\Phi_{\mathbf{x}}^{r}(\Phi_{\mathbf{x}}^{r\top}\Phi_{\mathbf{x}}^{r})^{-1} = QR^{-\top}$.

It is impossible to store the parameter-Jacobian $\Phi_{\mathbf{x}}^{m} \in \mathbb{R}^{N\times m}$ for very large models. It is also infeasible to compute and store each of the entries $\phi^{m}(\mathbf{x}_i)^{\top}\phi^{m}(\mathbf{x}_j)$ of the Gram matrix $\Phi_{\mathbf{x}}^{m}\Phi_{\mathbf{x}}^{m\top} \in \mathbb{R}^{N\times N}$. To circumvent this, we follow Park et al. (2023): Consider a random matrix $P \sim \mathcal{N}(0, q^{-1})^{m\times q}$ with $q \ll m$. This preserves the inner product with high probability,

$$\Phi_{\mathbf{x}}^{m}PP^{\top}\Phi_{\mathbf{x}}^{m\top} \approx \Phi_{\mathbf{x}}^{m}\Phi_{\mathbf{x}}^{m\top}. \tag{88}$$

Using this we may compute the Jacobian for minibatches out of $N$, and directly store the result in $\Phi_{\mathbf{x}}^{m}P \in \mathbb{R}^{N\times q}$. We use the highly optimized implementation by Park et al. (2023) that generates the projection coefficients of $P \in \mathbb{R}^{m\times q}$ as needed instead of storing them. This shows how to compute $\left(\Phi_{\mathbf{x}}^{r}(\Phi_{\mathbf{x}}^{r\top}\Phi_{\mathbf{x}}^{r})^{-1}\right)^{\top}\Phi_{\mathbf{x}}^{m}P \in \mathbb{R}^{r\times q}$, and thus approximate $B^{\top}B \in \mathbb{R}^{r\times r}$. Given $B^{\top}B$, finding the Cholesky decomposition $L \in \mathbb{R}^{r\times r}$ is $O(r^3)$. This is a one-time implementation and does not need to be repeated for every test point.

## D. Further Details on the Experimental Setup

### D.1. Regression

For each dataset and seed, we use a fixed three-way split of $72\%/18\%/10\%$ into train/validation/test. Inputs are standardized using the training-set mean and standard deviation, and targets are centered by subtracting the training set mean. The same split and preprocessing are used across all methods to ensure comparability.

All methods that rely on a neural network backbone use the same MLP architecture for regression: two hidden layers with width 50, ReLU activations, and a scalar output. They are trained using mean-squared error loss with a learning rate of $10^{-3}$ and gradient clipping of 1 with the AdamW optimizer, adapted to $\mu$P. The maximum number of training epochs is dataset-specific (*Boston*/*Concrete*/*Power*: 3000; *Energy*: 2000; *Wine*: 1000). Batch size is 32 for all datasets except for *Power*, where we use batch size 256. Model selection is performed based on validation RMSE, evaluated every 10 epochs. After selecting the optimal number of epochs, the backbone is retrained from scratch on the combined train and validation sets for exactly that number of epochs, and final results are reported on the held-out test set.

We treat predictive uncertainty as a Gaussian with mean from the backbone and variance from the post-hoc method. When a method requires the observation-noise variance, we either estimate it by validation-set MSE (empirical noise) or sweep a small grid and pick the best validation NLL.

Our method uses a fixed, trained backbone and constructs a feature-space Gaussian posterior. We form two feature sets at the training inputs: (i) last-layer (NNGP/BLL) features and (ii) all-but-last-layer Jacobian features. We fit a linear feature transform that best expresses the latter in the span of the former, and use it to define transformed features $L^\top \phi^r(x)$ that approximate the empirical NTK. We then perform standard Bayesian linear regression in feature space using these transformed features, yielding a predictive covariance that captures epistemic uncertainty. The subsampled variant builds the feature transform and posterior from a random $40\%$ subset of the training set (fixed per dataset and seed).

## D.2. Contextual Bandits

We use the *Wheel* Bandit benchmark with 2-D contexts sampled uniformly from the unit disk and 5 actions. Difficulty is controlled by ($\delta \in 0.5, 0.7, 0.9, 0.95, 0.99$). Rewards are Gaussian with fixed variance, and each run consists of alternating phases of environment interaction and model updates. All neural methods use the same action-conditioned MLP: inputs are the observation $x$ and the one-hot encoded action. They consist of two hidden layers of width 100 with ReLU, scalar output, and $\mu$-parameterization.

We train with MSE using the AdamW optimizer, learning rate $3 \times 10^{-3}$, and gradient clipping at 1.0. Each update phase uses batch size 512, with 100 gradient steps per update and 20 environment steps per phase, for a total of 80,000 gradient steps (16,000 environment steps). We initialize with 3 warm-start pulls per action before applying Thompson sampling. Thompson sampling is used throughout all the experiments. For each action, we sample a reward from its predictive distribution and select the action with the highest sample. Simple regret is evaluated using the posterior mean.

We maintain a feature-space Gaussian posterior derived from the fixed backbone. The posterior is rebuilt at a regular cadence (every gradient step in the default setting), and we optionally subsample the data used to build the posterior (i.e., when we use Rich-BLL (S)). When subsampling, we ensure the subset size is at least the feature dimension and rescale second-moment statistics to match the full-data scale. We also support an empirical noise option that updates the observation-noise variance online using a moving window of recent squared prediction errors; this updated variance is then used in posterior updates.

## D.3. Image Classification

We create a held-out validation split from the *CIFAR-10* training set and use a test set for evaluation. For OOD, we use *SVHN* and *CIFAR-100* test sets as out-of-distribution datasets. Input images are normalized with standard dataset statistics.

All post-hoc methods share the same backbone architecture and training protocol per run. We use a CNN with seven $3 \times 3$ convolutions, batch normalization and ReLU after each layer, two $2 \times 2$ max-pooling layers, global average pooling, and a final linear readout. With base width 128, the channel dimensions are $3 \rightarrow 128 \rightarrow 128 \rightarrow 256 \rightarrow 256 \rightarrow 256 \rightarrow 512 \rightarrow 512$, with pooling applied after the second and fifth convolutions. The final 512-dimensional feature vector is passed through dropout with probability 0.2 before the classifier. All convolutions use padding 1 and no bias terms. This network has approximately 5M trainable parameters. The CNN is trained with cross-entropy, $\mu$-parameterization, and the AdamW optimizer (learning rate $10^{-3}$ and no weight decay), for 100 epochs, with batch size 128 in training and 256 for evaluation, and a $10\%$ validation split. Further, training uses random crop and horizontal flips. We select the best checkpoint by validation accuracy and reuse the same trained backbone for all uncertainty methods to ensure fair comparison. We compute uncertainty at the logit level using a feature-space Gaussian posterior. Penultimate features are extracted from the backbone, and we build a feature transform that aligns these features to logit-level gradient features. Importantly, to build this transformation in a scalable way, we use the algorithm described in Appendix C. At test time we select the predicted class (or the true class for analysis), compute the logit-level predictive variance via feature-space GP inference, and use this

variance as the OOD score. We also evaluate an NNGP variant that uses penultimate features without the transformation, and a last-layer Laplace baseline with a shared precision. For Rich-BLL/NNGP/Laplace, we use a fixed noise variance of $0.1$, per-class feature subsets of size $1024$. For this experiment, we use ridge ($\lambda = 1.0$) to learn the transformation matrix, and the Laplace baseline uses ($\alpha = 1.0$) with softmax weighting and no clamp. Predictive NLL and ECE are computed via Monte-Carlo averaging with $50$ samples.

## E. Subsample Size Ablation

*Table 6.* Test Gaussian negative log-likelihood for Rich-BLL (S) under different subsampling ratios. The percentages indicate the portion of the dataset that was used for estimating the transformation matrix and for the posterior inference.

| Data used | *Boston* NLL ($\downarrow$) | *Concrete* NLL ($\downarrow$) | *Energy* NLL ($\downarrow$) | *Power* NLL ($\downarrow$) | *Wine* NLL ($\downarrow$) |
|---|---|---|---|---|---|
| 30% | $2.83 \pm 0.08$ | $3.10 \pm 0.03$ | $0.79 \pm 0.04$ | $2.78 \pm 0.01$ | $1.03 \pm 0.01$ |
| 50% | $2.79 \pm 0.08$ | $3.10 \pm 0.03$ | $0.78 \pm 0.04$ | $2.78 \pm 0.01$ | $1.03 \pm 0.01$ |
| 60% | $2.79 \pm 0.08$ | $3.10 \pm 0.03$ | $0.78 \pm 0.04$ | $2.78 \pm 0.01$ | $1.03 \pm 0.01$ |
| 70% | $2.79 \pm 0.08$ | $3.10 \pm 0.03$ | $0.78 \pm 0.04$ | $2.78 \pm 0.01$ | $1.02 \pm 0.01$ |
| 80% | $2.79 \pm 0.11$ | $3.10 \pm 0.03$ | $0.78 \pm 0.04$ | $2.78 \pm 0.01$ | $1.01 \pm 0.01$ |
| 90% | $2.79 \pm 0.12$ | $3.10 \pm 0.03$ | $0.78 \pm 0.04$ | $2.78 \pm 0.01$ | $1.01 \pm 0.01$ |

