# OpenReview forum: "Richer Bayesian Last Layers with Subsampled NTK Features"
_ICML.cc/2026/Conference — ICML 2026 regular_

### Official Review · Reviewer_b2Vp · 2026-03-12

**Soundness:** 2
**Presentation:** 3
**Significance:** 2
**Originality:** 2
**Overall Recommendation:** 4
**Confidence:** 4

**Summary:**

The paper provides an principled approximation to the estimate of Uncertainty for Neural network via inclusion of additional uncertainty that is ignored in the Bayesian Last layer method. To address this, the authors propose Rich-BLL, a method that approximates NTK-GP uncertainty while retaining the computational cost of standard Bayesian last layer inference. The key idea is to approximate earlier layer NTK features through a low-dimensional linear approximation derived from the last layer feature space. The paper’s main contributions are: proposing this approximation and its subsampled variant, showing that the resulting posterior variance is at least as large as that of a standard Bayesian last layer, deriving approximation bounds for the proposed estimate and subsampling variant, and empirically evaluating the method on regression, contextual bandits, image classification, and out-of-distribution detection tasks.

**Compliance With Llm Reviewing Policy:**

Affirmed.

**Final Justification:**

The authors answered my questioned promised to add further relevant results and simulations.

**Key Questions For Authors:**

my major concerns are as described in points 1, 2, 4 and 6 in the Strengths And Weaknesses section.

**Limitations:**

yes.

**Strengths And Weaknesses:**

The papers main contribution is the algorithm and study of its theoretical properties. Simulation/Empirical study is focused on toy examples. Thus I will focus my review of the main algo and theory. The theory seems technically straightforward and mostly an application of standard concentration arguments.

My main concerns are:
	1. How plausible is the linear assumption between \phi^m and  \phi^r. There should be some empirical study of this assumption. Otherwise the paper is exchanging computational cost for large bias in the variance estimate.
	2. The paper is also missing any study of the misspecification error. How does the error in the uncertainty propagate when $E(\phi^m|\phi^r) - A\phi^r$ is `large`?
	3. The claim that S^{bll} is an underestimate requires citation. Since that's the main motivation of the paper, this needs to be strongly motivated by citation or empirical results.
	4. Theorem 3.3 shows that that the proposed method leads to larger estimate than Bayesian Last Layer but doesn't talk about how this estimate compare to the full NTK estimate. Ideally we would like to understand the gap between S^{FULL NTK}, S^B, and S^{bll}.
	5. In Theorem 3.5 its said that "his bound is remarkable, as it does not grow with the number of training points N". I am not sure why that would be the case since 1/k (the subsample size) is the dominating term (as shown in the proof). I think this is really a mis-interpretation of the result. Concentration around A is not novel
	6. The more interesting result is Theorem A.4, which shows that approximation breaks down dramatically when the linear assumption is not correct. To convince us that the series of approximations proposed in the paper are useful and not just a set of tools to exchange computational cost for performance, the authors need to show us that these assumptions (at least empirically) are reasonable on real scale data.
Minor notes:
	a. Table 1, N, r, k and p are not defined yet.
	b. after (4), define r.
	c. a^{l+1} seems to be a mis-notation?
	d. Section A.3 paper argues informally that "these gradients are O(1) under NTK- or \mu-parameterization doesn't grow with m+r", this requires evidence or citation.

---

> ### Author Rebuttal · Authors · 2026-03-31
>
> We thank the reviewer for the careful reading and constructive feedback, and for highlighting several important theoretical and empirical aspects of our work. We try addressing all concerns below.
>
> ---
>
> > linear assumption
>
> ➡️ We address this concern in two ways.
>
> 1) We add a new empirical analysis of the spectrum of the empirical NTK for the models used in our experiments. We compute the NTK Gram matrix and report eigenvalue decay and the number of components capturing 90% of the trace. In all cases, we observe strong spectral concentration, indicating that the NTK has low effective rank.
>
> [See new plots](https://anonymous.4open.science/r/to-be-deleted-2-548F/MLP_eigendecay.jpg)
>
> 2) We add a new lemma B.1 (new App B) giving sufficient conditions under which the linear mapping between $\phi^m$ and $\phi^r$ is exact on the data. If the NTK has rank ≤ r, the network is in the NTK regime, and last-layer features are full rank, then NTK features lie in their span, i.e., $\phi^m(x)=A\phi^r(x)$.
>
> [See new lemma](https://anonymous.4open.science/r/to-be-deleted-2-548F/new_lemma.png)
>
> The empirical spectral decay shows that the NTK is effectively low-rank in practice, while the lemma establishes that in the low-rank regime the linear mapping is exact.
>
> ---
>
> > misspecification error
>
> ➡️  We do not explicitly study misspecification error. We follow a standard experimental protocol and evaluate across multiple datasets and settings (including regression, OOD detection, and bandits). If misspecification introduced large errors in the uncertainty estimates, it would degrade NLL, calibration, or downstream performance. Empirically, we do not observe such degradation.
>
> ---
>
> > S^{bll} is an underestimate requires citation
>
> ➡️ The statement that $S^{\text{BLL}}$ underestimates uncertainty is with respect to the NTK-GP, which we take as the “gold standard” in our analysis. In this setting, the result follows directly from the structure of the kernel: BLL corresponds to using only the last-layer features, our construction incorporates additional components arising from earlier layers, and the NTK-GP uses the full feature map of the network. As a result, the corresponding kernel matrix (and hence posterior covariance) is larger in the PSD sense, which is formalized in Theorem 3.3. We'll add a discussion deriving this relationship in the eNTK regime.
>
> Empirically, this behavior is also reflected in improved uncertainty quality (e.g., NLL, OOD detection), consistent with BLL underestimating uncertainty relative to richer feature representations.
>
> ---
>
> > Theorem 3.3 doesn't talk about full NTK
>
> ➡️ Our primary goal is to improve over BLL while retaining its computational cost, and we therefore focus the analysis on this comparison. The comparison to the full NTK-GP is mainly empirical: in simple settings we include direct comparisons (Figures 1-2), while in larger models computing the full NTK posterior becomes prohibitively expensive.
>
> The gap between $S^{\text{FULL NTK}}$ and $S^B$ depends on the quality of the feature approximation. In the case where the approximation is exact (we provide sufficient conditions for this in the new lemma B.1 in Appendix B), the two coincide, i.e., $S^{\text{FULL NTK}} = S^B$. Outside this regime, the gap depends on approximation quality and is hard to characterize in closed form.
>
> We will emphasize that our method can be viewed as a tractable approximation to the full NTK-GP, with equality in the exact low-rank regime in the revision.
>
> ---
>
> > Theorem 3.5 "this bound is remarkable"
>
> ➡️ We agree that the interpretation of Theorem 3.5 can be clarified and will revise the text accordingly. The key point is that the bound depends on the subsample size $k$ rather than the full dataset size $N$, but we acknowledge that the current phrasing may be misleading.
>
> ---
>
> > more interesting result is Theorem A.4
>
> ➡️ Similar to our response to 1., we now added a new empirical analysis of the NTK spectrum for the models used in our experiments. The results show strong spectral concentration (low effective rank) across all models used in our experiments, indicating that the feature space is well-approximated by a low-dimensional subspace. In this regime, the linear approximation is expected to be reasonably accurate.
>
> In addition, our empirical results show consistently strong uncertainty quality (NLL, OOD, bandits), which would be sensitive to the type of misspecification highlighted by Theorem A.4. This provides further evidence that the assumptions underlying the method are reasonable in the settings we consider.
>
> ---
>
> > Minor notes
>
> a) We have now defined $N$, $r$, $k$, and $p$ in the caption of Table 1.
> b) We now define $r$ in the sentence following Eq. (4).
> c) Typo fixed: $a^l$ replaces $a^{l+1}$.
> d) We have added appropriate citations to support the gradient scaling under NTK- and $\mu$p.
>
> ---
>
> We hope this addresses your concerns and kindly ask you to reconsider your score. Happy to clarify further.

---

> > ### Author Rebuttal · Reviewer_b2Vp · 2026-04-03
> >
> > increased the rating to 4. Thanks for the response.

---

> > > ### Author Response · Authors · 2026-04-07
> > >
> > > Thank you for your thorough review and for engaging carefully with our responses. We especially appreciate your feedback on the theoretical assumptions and empirical validation, which helped us strengthen both aspects of the paper. We are grateful for your updated positive assessment.

---

### Official Review · Reviewer_WAPX · 2026-03-13

**Soundness:** 1
**Presentation:** 2
**Significance:** 2
**Originality:** 2
**Overall Recommendation:** 2
**Confidence:** 4

**Summary:**

This paper considers Bayesian Uncertainty Quantification for Neural Networks. Specifically, it seeks to improve the performance of last-layer Bayesian methods, those that first linearize a network around its final-connected weights, and then form a Bayesian generalized linear model from the resultant model. To achieve this, the authors conjecture that there exists a linear transformation between the last-layer parameter gradient and the full parameter gradient. Through this transformation, they can approximate the full predictive covariance with a cost that scales cubically with final layer dimension. The authors present the mathematical derivation of this approximate covariance, as well as a theorem comparing the ordering of the approximate vs. last-layer covariance, and results on the convergence rate for forming the linear transformation. A series of small-scale experiments are conducted, in which the proposed method performs comparably against competing methods.

**Compliance With Llm Reviewing Policy:**

Affirmed.

**Final Justification:**

The authors have addressed a number of concerns that I had with the paper, however a few issues remain. The most prominent of these is the assumption that the last layer feature map has rank at least as large as the eNTK. While it is true that the eNTK has a significant proportion of very small eigenvalues (which was demonstrated by the authors in terms of the trace), there is no reason to believe that the rank of the last layer approximation is anywhere close to this effective rank. In fact, the last layer matrix is likely to 'suffer' from low effective rank as well, and it has not been demonstrated that this is not the case. The theoretical contributions, and the method itself, are all predicated on this assumption, and while this would not be an obstruction in light of strong numerical evidence, in my view this not been demonstrated empirically. There also appears to be a misunderstanding in the result of He. This paper demonstrated that the NTK-GP is in fact **not** a reasonable "ground truth" for infinite width networks, and so the claim that the authors provide a more efficient implementation of He's theory is questionable.

For these reasons, I have retained my score. I cannot recommend it for acceptance at this stage. I would like to emphasize that I think the core idea of this paper is nice, however its execution is currently is lacking. This paper would benefit from more detailed revision than the review period allows, and I hope to see an updated version of this paper in the future.

**Key Questions For Authors:**

Can you please address the issues raised in the strengths and weaknesses section?

**Limitations:**

No discussion of limitations. Perhaps consider how this would perform for classification, if one can check the applicability of the linear transformation or why not, if this method could be expected to give *worse* performance in some instances, etc.

**Strengths And Weaknesses:**

**Soundness:**

Strengths:
- Most proofs are clear and rigorous. Except for the proof for Theorem 3.3., they seem to be correct.
- Experimental results include mostly appropriate competing methods.
- Inclusion of bandit problem is appreciated.

Weaknesses:
- Leading motivation for this work is that use of NTK-GPs "requires solving a linear system whose dimension grows with the number of training points"; the work by Wilson 2025 is referenced here. However, this work specifically sidesteps this requirement, instead sampling from an NTK-GP using SGD, yet there is no discussion of this.
- Motivation for use of low-rank approximation lacking. Three papers [1, 2, 3] are used to motivate idea of eigen-decay, but [1] pertains to deep ResNets (of which none are featured in the experiments), [2] seems to consider the effect of different activation function on eigen-decay, and [3] seems to require infinite-width. Further, it is not obvious why the eigen-decay means that the purported low-rank approximation is the final-features. Could this low-rank subspace not be separate from the final-layer space entirely? A more in-depth argument for this would greatly strengthen the paper.
- Theorem 3.3 states that using the linear transformation gives higher uncertainty over all test points. Firstly, it is not clear that this is always desired (some times you may want small uncertainty for a test point, so a poor lower-bound would hinder the approximation), and secondly, the attached proof does not appear to demonstrate the result. Even using the resolvent identity I could not see it. Clarification of the proof would be highly valuable.
- Issue with experiments. Dropout often gives better results than DE in classification experiments, though this goes against most published UQ results [4, 5]. For the regression setting, the MAP has a NLL score, which requires a computed variance value. Where does this come from for the MAP (which is stated to have a scalar output)? No code was provided, so results cannot be easily validated. The CNN is not defined, no details of test accuracy are given (this is an issue, as a LeNet performs alot differently than a ResNet50). A lot of claims are made about scalability, yet networks used for validating this claim are quite small. Experimental procedure for classification completely avoids between-class covariance, as single class is taken before uncertainty is computed; this would be expected to hinder performance.
-  Some quoted papers are incorrectly quoted; i.e. reference to [6]. This work contains no mention of KFAC, Hessian, Fisher etc.

**Presentation:**

Strengths:
- Proofs are clear and well structured (with the exception of proof to Theorem 3.3).
Weaknesses:
- Bayesian UQ should be introduced, i.e. idea of posterior/predictive dist.
- Some incorrect explanations of important topics, e.g. "The Neural Tangent Kernel (NTK) (Jacot et al., 2018) captures this effect for **wide** neural networks" (NTK-GPs have been used for UQ, but they do not require wide networks), and "A first-order Taylor expansion of the network around $\theta$ motivates representing the effect of small parameter perturbations through the network’s parameter gradients" (this isn't the motivation), discussion of last-layer refinement methods seems incorrect (refinement should mean improvement, but things such as LLA and VI are just further approximations on the last-layer approximation).
- Section 2 and 3 should be written with more clarity and rigor. I.e., what does 'optimal parameters' mean, define the network dimensions, variable $y$ is obviously output, but this is never stated. Use 'vec' to state the concatenation of two vectors.
- Name of method (RichBLL) is never actually introduced; rather, authors just begin to use it in the experiments.

**Significance:**

Strengths:
- Scalability of Bayesian methods is an important area of research, and low-rank approximation of NTK seems a promising avenue to consider.
- Useful extension of [7] to UQ.
- Appendix B uses a result from [8, 9], for forming the NTK using random projections. This could be very useful to the UQ community.

Weaknesses:
- Final two theoretical results have limited significance; Theorem 3.4 and Theorem 3.5 are simply a $n^{-0.5}$ convergence of least-squares type result. They do not claim that the error in the low-rank approximation goes to zero; rather it is simply a law-of-large numbers result, i.e. the sample-mean estimate versus the expected value. Further, constant in bound of Theorem 3.5 may be very large ($\lambda_{\min}$ may be tiny, and $K$ may be massive).
- Weaknesses in experimental methodology reduces significance of experimental results.
- No discussion of applicability to classification.

**Originality**
Strengths:
- Novel amalgamation of previous ideas to the context of UQ.

Weaknesses:
- Motivation for introducing these previous ideas in the context of UQ needs improvement. I.e., use of random projections should be brought to the main body, and idea of linear transformation should be better motivated.

References:

[1] Belfer, et. al. "Spectral analysis of the neural tangent kernel for deep residual networks", 2024.

[2] Murray, et. al. "Characterizing the spectrum of the ntk via a power series expansion", 2022.

[3] Bowman, "On the spectral bias of neural networks in the neural tangent kernel regime", 2023.

[4] Maddox, et. al. "A simple baseline for Bayesian uncertainty in deep learning", 2019.

[5] Lakshminarayanan et. al. "Simple and Scalable Predictive Uncertainty Estimation using Deep Ensembles", 2017.

[6] Mukhoti, et. al. "Deep deterministic uncertainty: A new simple baseline", 2023.

[7] Ciosek, et. al. "Linear gradient prediction with control variates", 2025.

[8] Park, et. al. "Attributing model behavior at scale", 2023.

[9] Johnson, et. al. "Extensions of Lipschitz mappings into a Hilbert space", 1984.

---

> ### Author Rebuttal · Authors · 2026-03-31
>
> We thank the reviewer for the feedback and for highlighting both strengths and areas for clarification.
>
> ---
>
> > Wilson 2025
>
> ➡️ Wilson et al. approximate the NTK posterior via sampled linearized models, requiring multiple optimization runs and test-time passes. In contrast, our method yields a closed-form posterior in a low-dimensional feature space with BLL-level inference cost. They rely on a full row-rank Jacobian assumption, whereas our approach leverages the empirically observed low effective rank of the NTK. We will add this in the related work
>
> ---
>
> > Motivation for use of low-rank
>
> ➡️ We strengthen this motivation empirically and theoretically (see new plots).
>
> - Empirically, we observe strong spectral concentration: for UCI, ~50–60 eigenvalues explain 90% of the trace (2000 points), and for CIFAR-10, ~80–100 suffice, indicating a low-dim structure
>
> [See new plots](https://anonymous.4open.science/r/to-be-deleted-2-548F/MLP_eigendecay.jpg)
>
> - Eigenvalue decay alone doesn't guarantee structure is captured by last-layer features. We add a lemma giving sufficient conditions for exactness: if the eNTK feature matrix has rank ≤ r and Φ_x^r has full column rank, then Φ_x^m = Φ_x^r A^⊤ for some A.
>
> [See new lemma](https://anonymous.4open.science/r/to-be-deleted-2-548F/new_lemma.png)
>
> ---
>
> > Theorem 3.3
>
> ➡️ The result is relative to BLL, which ignores earlier-layer uncertainty. Our method adds part of this back, increasing predictive covariance. In well-sampled regions both methods concentrate, while in low-data regimes the additional uncertainty is beneficial (as seen in our results)
>
> The proof follows from a PSD ordering argument. Since BᵀB = AᵀA + I_r ⪰ I_r, we have (BᵀB)⁻¹ ⪯ I_r, which implies (1/σ² Φ_x^{rᵀ}Φ_x^r + (BᵀB)⁻¹)⁻¹ ⪰ (1/σ² Φ_x^{rᵀ}Φ_x^r + I_r)⁻¹. Multiplying by Φ_{x’}^r preserves the order, yielding S^B ⪰ S^{BLL}.
>
> ---
>
> > Issue with experiments
>
> (i) **Dropout vs Ensembles.** Corrected a typo in Table 2: ensembles outperform dropout on Concrete. For img classification, increasing ensemble size (4→7) outperforms dropout (NLL 0.29, SVHN AUROC 0.93)
>
> (ii) **NLL for MAP**. We use a Gaussian predictive distribution with mean from the network and variance from validation MSE.
>
> (iii) **Reproducibility**. We will release the full codebase upon acceptance
>
> (iv) **CNN**. We now include full details in Appendix C.3
>
> (v) **Between-class covariance**. We evaluate per-logit uncertainty ignoring cross-class covariance (like common baselines, e.g., mean-field methods) and perform well empirically
>
> ---
>
> > papers incorrectly quoted
>
> ➡️ We thank the reviewer; the citation will be corrected
>
> ---
>
> > incorrect explanations
>
> ➡️ We respectfully disagree with the reviewer’s first point. The NTK corresponds to the infinite-width regime where the GP interpretation is exact; for finite networks, the eNTK is a linearized approximation. Linearized Laplace uses these features but operates outside this regime. To our knowledge, no prior work demonstrates effective NTK-GP-based UQ in shallow/narrow networks. Our method targets practical finite-width settings while leveraging eNTK structure
>
> The sentence on the first-order Taylor expansion refers to the standard derivation of empirical NTK features: linearization around θ̂ yields the Jacobian feature map. Finally, we will revise the wording around “refinement” to make sure it is clear
>
> ---
>
> > Clarifications
>
> ➡️ We will expand Section 2 to briefly introduce Bayesian UQ. We will clarify that “optimal parameters” refer to the converged solution (i.e., MAP), that the theoretical analysis assumes scalar outputs wlog, and that $y$ is the output. We will standardize notation ($\mathrm{vec}$) and introduce “Rich-BLL” in Sections 1 and 3
>
> ---
>
> > Final two theoretical results
>
> ➡️ While Theorems 3.4 and 3.5 are concentration results, they converge to the population NTK projection. When the NTK is low-rank, this recovers the true feature subspace and yields correct uncertainty estimates. Thus, it is convergence to the correct object
>
> Theorem 3.3 guarantees that our predictive uncertainty dominates that of BLL, ensuring that our method captures additional uncertainty beyond the last layer. Finally, we note that the constants in Thrm 3.5 can be large in ill-conditioned regimes; the result is intended to capture qualitative scaling with subsample size rather than tight bounds
>
> ---
>
> > Classification
>
> ➡️ We include both a classification methodology and experiments (Sec 4.4), including calibration and OOD metrics
>
> ---
>
> > Motivation introducing previous ideas
>
> ➡️ We will move the use of random projections (for efficiently approximating gradient inner products) from App B to the main body
>
> Motivation for the linear transformation follows directly from the low-rank NTK structure discussed above: it approximates the full eNTK feature map while retaining BLL cost, and is exact under the conditions of the new lemma
>
> ---
>
> We hope this addresses your concerns and kindly ask you to reconsider your score

---

> > ### Author Rebuttal · Reviewer_WAPX · 2026-04-04
> >
> > Thanks for your quick response. There are still several issues for which I'd appreciate some clarification.
> >
> > **Motivation for use of low-rank:**
> > While I appreciate the new theoretical result, unfortunately the attached lemma does not answer my question; the assumption in the attached lemma is much too strong, as assuming that all the structure in the full Jacobian (rank r) arises from the last-layer features (rank r) makes the result immediate. My question is rather the following: if the full-Jacobian is essentially low-rank, how do we know that the low-rank structure corresponds to the last-layer, instead of say structure from previous layers, or some manifold related to data structure?
> >
> > **Experimental Notes:**
> > I still have several concerns regarding the experiments. Firstly, for the NLL for MAP, how can a single, scalar output NN return a variance? Secondly, releasing the codebase during the rebuttal would be *highly* beneficial for validation of results. Further, could you provide the structure and size of the CNN during the rebuttal? Finally, mean-field methods are not SOTA in the literature, and most SOTA/adjacent methods do consider between class-covariance (DE, SWAG, SNGP, etc., though with considerable exception of LLA). Independent outputs is a large approximation to make.
> >
> > **Incorrect Explanations:**
> > Note that the infinite-width NTK theory does not give an exact GP interpretation (see [1]), and for finite-width networks, LLA [2] / NUQLS [3] returns an eNTK-GP posterior. Hence, I believe the text should be amended to reflect this. Regarding the statement on Taylor expansions, I believe if you were referring to the derivation of eNTK features, this sentence needs to be reworded for clarification.
> >
> > **Final two theoretical results:**
> > The significance of these results only holds if the low-rank structure of the NTK corresponds to the last-layer. If this is not the case (see response to low-rank motivation), then you may be measuring convergence to a sub-optimal approximation.
> >
> > **Conclusion:**
> > As it stands, my concerns for this work still remain, specifically and most critically the justification for a linear transformation between the full- and last-layer Jacobians, the lack of experimental rigour, as well as some other more minor concerns, as listed above. As it stands, I will keep my current score, but I look forward to hearing your final response.
> >
> > **References:**
> >
> > [1] He, et. al. "Bayesian Deep Ensembles via the Neural Tangent Kernel", NeurIPS 2020.
> >
> > [2] Immer, et. al. "Improving predictions of Bayesian neural nets via local linearization", NeurIPS 2020.
> >
> > [3] Wilson, et. al. "Uncertainty Quantification with the Empirical Neural Tangent Kernel", NeurIPS 2025.

---

> > > ### Author Response · Authors · 2026-04-05
> > >
> > > > Motivation for use of low rank
> > >
> > > ➡️ We understand the reviewer’s concern and augment the appendix with a detailed analysis based on a relative quasi low-rank notion. Instead of assuming the NTK is exactly rank $r$, we measure how far the full feature matrix lies from the span of the last-layer features via a projection residual $\varepsilon_{\mathbf{x}}$. We show that the least-squares map used in our method is optimal in this sense: it achieves exactly this residual, i.e., the approximation error equals the distance to the last-layer span. **Thus, we do not assume the low-rank structure comes from the last layer, but quantify how well its span captures the dominant structure of the full Jacobian.**
> > >
> > > We further show that this **feature-level error propagates in a controlled way to the predictive covariance**, with posterior error bounded by the training residual $\varepsilon_{\mathbf{x}}$ and test residual $\varepsilon_{\mathbf{x}’}$. Hence, if the dominant NTK subspace aligns well with last-layer features, the approximation is accurate; otherwise, the error is explicitly quantified.
> > >
> > > Finally, our empirical spectral analysis shows strong eigenvalue decay, with most of the NTK trace explained by few directions (i.e., effectively low-rank), supporting that the method operates in a regime where small residuals are reasonable.
> > >
> > > [An extended discussion which is now included in the revised paper can be found here](https://anonymous.4open.science/r/to-be-deleted-2-548F/quasi-low-rank.pdf)
> > >
> > > ---
> > >
> > > > Experimental Notes
> > >
> > > 1. For the MAP, we assume a Gaussian predictive distribution with mean given by the network output and a **constant variance estimated from the MSE on a val set**. This allows us to compute NLL in a consistent way across methods and is standard practice.
> > >
> > > 2. We cannot release the codebase now because we face legal constraints (we will release it once the paper is accepted). But  for transparency, we share a notebook tutorial of our method (benchmarking BLL with and without subsampling, and the MAP on the UCI Power dataset for clarification of 1).
> > >
> > > [Illustrative notebook](https://anonymous.4open.science/r/to-be-deleted-2-548F/uci_regression_tutorial-anon.ipynb)
> > >
> > > 3. What we will add to our revised appendix: We use a CNN with seven 3×3 convolutions, batch normalization and ReLU after each layer, two 2×2 max-pooling layers, global average pooling, and a final linear readout. With base width 128, the channel dimensions are 3→128→128→256→256→256→512→512, with pooling applied after the second and fifth convolutions. The final 512-dimensional feature vector is passed through dropout with probability 0.2 before the classifier. All convolutions use padding 1 and no bias terms. This network has approximately 5M trainable parameters.
> > >
> > > 4. We apologise for the confusion introduced when mean-field approximations were raised. We don't model classification as independent classwise posteriors. Instead, for each input we select a single decision-relevant scalar score from the logits, by default the predicted logit $f_{c^*}(x)$ with $c^*=\arg\max_k f_k(x)$ and apply our regression-style approximation only to that scalar. Unlike mean-field multi-class methods, which define separate latent variables for all class outputs and factorize across them, we do not define a factorized multi-output posterior at all. Hence, **we do not make the assumption of independent outputs**. Our empirical evidence is good indicating a sensible modelling choice.
> > >
> > > ---
> > >
> > > > Incorrect Explanations
> > >
> > > ➡️ We will edit the paper for clarity. For wide networks, the ground truth for estimating the posterior is given by plugging the NTK into the GP posterior formula. The question is how to compute that posterior efficiently. He et al. provide a way of doing this by training a version of the ensemble (compute cost scales with the number of networks in the ensemble), we do it in a way that only requires one network (by making a different approximation).
> > >
> > > ---
> > >
> > > > Final 2 theory results
> > >
> > > ➡️ We do maintain that, by linear algebra, the low-rank structure of the matrix is global in the sense that we can predict the full Jacobians from any rank-$r$ set of vectors, not necessarily the last-layer activations (although they are the most convenient).
> > >
> > > We now show a relaxation of our previous result, where a quasi-rank setup is considered (see link above). Note that the quasi-rank assumption is entirely verifiable empirically, hence there is no gap between theory and practice on this issue. We accept these issues are not explained well - we'll explain this better in the final paper.
> > >
> > > ---
> > >
> > > We believe we have addressed all concerns to the best of our ability through new empirical results, additional theoretical analysis, and clarifications, and that the paper is now in a significantly stronger position; given this and the positive assessments from the other reviewers, we kindly ask for an increase in the score and thank the reviewer for the productive rebuttal.

---

### Official Review · Reviewer_zXc4 · 2026-03-13

**Soundness:** 3
**Presentation:** 2
**Significance:** 3
**Originality:** 3
**Overall Recommendation:** 4
**Confidence:** 2

**Summary:**

This paper proposes Rich-BLL, a post-hoc uncertainty estimation method that augments Bayesian Last Layers (BLLs) with a data-driven correction derived from empirical NTK (eNTK) features. The key idea is to project non-last-layer Jacobian features onto the span of last-layer features via least squares, yielding a small r×r positive definite matrix that reweights last-layer features and leads to a closed-form posterior in r dimensions with the same inference cost as a standard BLL. The authors prove that Rich-BLL yields predictive variances that are always at least as large as BLL, provide subsampling schemes with finite-sample concentration bounds, and demonstrate improved calibration and exploration across UCI regression, contextual bandits, and CIFAR-10 OOD detection, with minimal overhead relative to BLL during inference.

**Compliance With Llm Reviewing Policy:**

Affirmed.

**Final Justification:**

Thank you to the authors for their response, I will keep my score.

**Key Questions For Authors:**

1. In classification, how exactly are predictive probabilities and NLL computed from the logit-level variance? Is a Gaussian–probit/softmax link used, or are scores only used for OOD? Please clarify to ensure fair comparison with baselines and report accuracy alongside Expected Calibration Error (ECE).

**Limitations:**

yes

**Strengths And Weaknesses:**

### Main strengths
- This paper introduces a principled, low-dimensional correction to last-layer Bayesian inference grounded in NTK feature projections, effectively importing some upstream-layer uncertainty into BLLs without scaling inference with dataset size or full parameter count.
- Core derivations (Theorems 3.1–3.3) are compact and transparent.
- The method is evaluated across diverse settings (UCI regression with NLL, contextual bandits measuring exploration via regret, and image OOD detection/calibration), supporting claims that it improves epistemic uncertainty over BLLs.

### Weaknesses
I do not see any major weaknesses that would invalidate the main contributions.

---

> ### Author Rebuttal · Authors · 2026-03-31
>
> We thank the reviewer for the positive and careful assessment of our work, and for clearly summarizing the main contributions and strengths. We also appreciate the constructive question regarding the classification setup.
>
> ---
>
> > In classification, how exactly are predictive probabilities and NLL computed from the logit-level variance?
>
> ➡️ We thank the reviewer for this question. As described in Appendix C.3, in our classification setup we model uncertainty at the logit level using a feature-space Gaussian posterior. Penultimate features are mapped to logit-level gradient features via the transformation in Appendix B, and at test time we compute the predictive variance of a selected logit (predicted or true class) using GP inference. This variance is used as an OOD score.
>
> We do not map this uncertainty to predictive probabilities (e.g., via a probit or softmax link). Instead, NLL and ECE are computed using the backbone softmax probabilities only, while variance is used solely for uncertainty-based metrics such as OOD detection.
>
> Regarding accuracy, our method is post-hoc and does not modify the predictive mean, so accuracy is identical to the backbone; we can include it in the table for completeness.
>
> ---
>
> We hope this clarification resolves the remaining question; if so, we would greatly appreciate a reconsideration of the score. We are of course happy to provide any further clarification if needed.

---

> > ### Author Rebuttal · Reviewer_zXc4 · 2026-04-02
> >
> > Thank you to the authors for their response, I will keep my score.

---

> > > ### Author Response · Authors · 2026-04-07
> > >
> > > Thank you for your careful review and for considering our responses. We appreciate your feedback on the experimental setup, which helped us clarify the classification experiments, and we are grateful for your continued positive evaluation.

---

### Decision · Program_Chairs · 2026-04-30

**Decision:**

Accept (regular)

**Comment:**

This paper proposed a method called Rich-BLL which offers improved uncertainty quantification relative to Bayesian Last Layer (BLL) approaches by incorporating an empirical NTK (eNTK) correction from the previous layers. Reviewers recognized the soundness of the paper, its theoretical core derivations, and the breadth of empirical results. During the rebuttal phase, one of the concerns focused on the assumption that the feature map of the final layer has rank at least as large as the eNTK, since the theoretical contributions rely on this assumption.  Although it is true that the authors have not proved that this assumption will hold, during the rebuttal they did provide evidence that eNTK has a significant proportion of small eigenvalues. They also added discussion characterizing the feature approximation error in cases where the full eNTK features lie outside the span of last-layer features. Therefore, this point has been sufficiently resolved and thus I recommend acceptance.